# Tyrosine phosphorylation controlled poly(A) polymerase I activity regulates general stress response in bacteria

Nimmy Francis[1], Malaya R Behera[1,2], Kathiresan Natarajan[3], Rakesh S Laishram[1]

**RNA 3′-end polyadenylation that marks transcripts for degradation is implicated in general stress response in *Escherichia coli*. Yet, the mechanism and regulation of poly(A) polymerase I (PAPI) in stress response are obscure. We show that *pcnB* (that encodes PAPI)-null mutation widely stabilises stress response mRNAs and imparts cellular tolerance to multiple stresses, whereas PAPI ectopic expression renders cells stress-sensitive. We demonstrate that there is a substantial loss of PAPI activity on stress exposure that functionally phenocopies *pcnB*-null mutation stabilising target mRNAs. We identify PAPI tyrosine phosphorylation at the 202 residue (Y202) that is enormously enhanced on stress exposure. This phosphorylation inhibits PAPI polyadenylation activity under stress. Consequentially, PAPI phosphodeficient mutation (tyrosine 202 to phenylalanine, Y202F) fails to stimulate mRNA expression rendering cells stress-sensitive. Bacterial tyrosine kinase Wzc phosphorylates PAPI-Y202 residue, and that *wzc*-null mutation renders cells stress-sensitive. Accordingly, *wzc*-null mutation has no effect on stress sensitivity in the presence of *pcnB*-null or *pcnB-Y202F* mutation. We also establish that PAPI phosphorylation-dependent stress tolerance mechanism is distinct and operates downstream of the primary stress regulator RpoS.**

## Introduction

Polyadenylation (addition of a polyadenosine tail, PA-tail) at the 3′-end is a major post-transcriptional RNA modification event in bacteria (Sarkar, 1997; Mohanty & Kushner, 2011; Hajnsdorf & Kaberdin, 2018). Polyadenylation marks RNAs for degradation and regulates RNA quality control in the cell (Steege, 2000; Dreyfus & Regnier, 2002). Poly(A) polymerase I (PAPI) is the primary enzyme that polyadenylates transcripts at the 3′-end in *Escherichia coli* (Liu & Parkinson, 1989; Cao & Sarkar, 1992; Raynal et al, 1996). Another protein, 3′-exonuclease polynucleotide phosphorylase (PNPase), also acts as a PAP that accounts for residual polyadenylation in the absence of PAPI and incorporation of non-adenosine nucleotide in

the 3′-end (Mohanty & Kushner, 1999b, 2000). PAPI largely adds short PA-tail that ranges from 10 to 40 adenosines at the 3′-end on most RNAs (O'Hara et al, 1995; Mohanty & Kushner, 1999a). However, there are no specific polyadenylation signals on bacterial RNAs, and PAPI largely targets transcripts with structured 3′-end that includes mRNAs with intrinsic transcription terminators (Blum et al, 1999; Khemici & Carpousis, 2004; Mildenhall et al, 2016). The addition of a PA-tail provides a platform for the 3′-exonuclease in the degradosome complex to initiate exonucleolytic degradation of structured RNA 3′-ends (Haugel-Nielsen et al, 1996; Coburn & Mackie, 1998; Blum et al, 1999; Coburn et al, 1999).

PAPI is a member of the nucleotidyltransferase superfamily (includes eukaryotic PAPs) that adds a 3′-ribopolymeric tail in a template-independent manner (Masters et al, 1990; Holm & Sander, 1995; Martin & Keller, 1996; Yue et al, 1996; Aravind & Koonin, 1999). PAPI is a 473–amino-acid-long monomeric protein with an N-terminally located catalytic domain, a C-terminally located RNA-binding domain, and a 17-residue amino-terminal leader sequence that is cleaved post-translationally (Raynal & Carpousis, 1999; Yehudai-Resheff & Schuster, 2000). The PAPI catalytic domain consists of three distinct motifs (M1, M2, and M3) with three active sites that form the catalytic triad and a nucleotide interaction region required for PAPI to function (Yehudai-Resheff & Schuster, 2000; Toh et al, 2011). Point mutation of the 205-aspartic acid (D205) residue or the 208-arginine (R208) residue in the nucleobase-interacting region abolishes PAPI activity (Raynal & Carpousis, 1999; Betat et al, 2004; Just et al, 2008). In addition, there are binding regions for the degradosome component proteins RNase E, RhlB, or HfQ that potentially affect PAPI function or activity (Raynal & Carpousis, 1999; Hajnsdorf & Regnier, 2000; Yehudai-Resheff & Schuster, 2000; Carabetta et al, 2010). PAPI enzyme is encoded by the *pcnB* gene that is transcribed from three σ70-dependent promoters (P1, P2, and PB) and two σS-dependent promoters (Cao & Sarkar, 1992; Xu et al, 1993; Jasiecki & Węgrzyn, 2006b). However, the two important σ70 (σD)-dependent promoters (P1 and P2) are primarily active under growth-limiting conditions, and the two σ38 (σS)-dependent promoters are inhibited by guanosine tetraphosphate (ppGpp) and the transcription factor DksA (Nadratowska-Wesolowska et al, 2010). In addition, a

[1]Cardiovascular and Diabetes Biology Group, Rajiv Gandhi Centre for Biotechnology, Trivandrum, India   [2]Regional Centre for Biotechnology, Faridabad, India   [3]Transdisciplinary Biology Program, Rajiv Gandhi Centre for Biotechnology, Trivandrum, India

Correspondence: laishram@rgcb.res.in

non-canonical initiation codon (AUU) and a poor ribosome-binding site regulate *pcnB* mRNA translation to maintain a low PAPI level in the cell (Binns & Masters, 2002). Apart from the transcriptional and translational control of *pcnB* gene expression, a post-translational phosphorylation of PAPI by a *Bacillus subtilis* serine kinase PrkC reduces PAPI activity in vitro (Jasiecki & Węgrzyn, 2006a). However, PAPI in vivo phosphorylation sites or its cellular significance remains to be defined.

PAPI was initially identified as a regulator of plasmid copy number via turnover of small non-coding RNA, *RNAI* (He et al, 1993; Xu et al, 1993; Cohen, 1995). Subsequent studies have shown PAPI to destabilise different RNA species including mRNAs, ribosomal RNAs, transfer RNAs, antisense RNAs, and small regulatory RNAs (Dam Mikkelsen & Gerdes, 1997; Söderbom et al, 1997; Li et al, 1998; Reichenbach et al, 2008; Maes et al, 2012, 2016; Mohanty et al, 2012). tRNA polyadenylation can limit the availability of charged tRNA for translation that in turn cease protein synthesis causing cellular toxicity (Mohanty & Kushner, 2012). Polyadenylation of small regulatory RNAs (GlmY, GcvB, SroH, RybB, RyjA, and GlmZ) regulates various cellular processes or growth conditions including DNA repair, iron homeostasis, amino acid metabolism, motility and chemotaxis, and adaptation to stress (Reichenbach et al, 2008; Maes et al, 2016; Ruiz-Larrabeiti et al, 2016; Holmqvist & Wagner, 2017; Sinha et al, 2018). mRNA polyadenylation also regulates functional gene expression in a repertoire of cellular functions (stationary phase growth, chemotaxis and motility, nutrient starvation, and envelope stress) (Aiso et al, 2005; Santos et al, 2006; Joanny et al, 2007; Reichenbach et al, 2008; Carabetta et al, 2010; Maes et al, 2012, 2013, 2016; Mohanty & Kushner, 2012; Sinha et al, 2018). However, the *pcnB* gene is non-essential in the cell and *pcnB*-null mutation exhibits no discernible growth defects under normal growth condition (Masters et al, 1993; Gerdes et al, 2003). Recent studies have indicated roles of RNA polyadenylation in general stress response in bacteria (Yamanaka & Inouye, 2001; Maes et al, 2013; Francis & Laishram, 2021). However, the mechanism of PAPI-mediated stress response, its interplay with the RpoS-mediated transcriptional pathway, or PAPI regulation during stress is unclear.

Genome-wide RNA-Seq analysis after *pcnB*-null mutation reveals stabilisation of primarily stress response mRNAs among the *pcnB* targets. Consequentially, *pcnB*-null mutation imparts cellular tolerance to multiple stresses, whereas PAPI ectopic expression renders cells stress-sensitive. Furthermore, to understand the mechanism of PAPI-mediated stress response, we investigated PAPI activity and *pcnB* mRNA expression on stress exposure. We show that there is >70% reduction in the PAPI activity on stress exposure stabilising target mRNAs but not the *pcnB* cellular expression level. We detected induced tyrosine (but not serine) phosphorylation on PAPI that inhibits PAPI polyadenylation activity under stress. We show that PAPI tyrosine phosphorylation at the catalytic domain (Y202) by bacterial tyrosine kinase Wzc reduces PAPI activity and stimulates stress response gene expression. We show that Y202 phosphorylation of PAPI induces structural alterations in the protein that affects ATP interaction. Subsequently, introduction of *wzc*-null mutation or PAPI phosphodeficient mutation (tyrosine 202 to phenylalanine, Y202F) renders cells stress-sensitive. However, *wzc*-null mutation has no effect on stress sensitivity in the presence of *pcnB*-null or *pcnB-Y202F* mutation. We also show that this PAPI

phosphorylation–dependent stress tolerance mechanism is distinct but operates downstream of the transcriptional induction by RpoS during stress.

# Results

## PAPI-mediated polyadenylation regulates general stress response gene program in *E. coli*

RNA-Seq analysis after *pcnB*-null mutation ($\Delta pcnB759::kan$) in MG1655 background (MG-*pcnB*) shows most (~170 genes) of the total mRNAs stabilised on the mutation (600 genes) were those involved in various stress response pathways (DNA damage, oxidative stress, osmotic stress, heat or cold shock, nutrient starvation, or biofilm formation) (Figs 1A–C and S1A and B). The total list of up-regulated and stress response genes stabilised on *pcnB*-null mutation is shown in Tables S1 and S2, respectively. Interestingly, the master regulator of stress response encoded by *rpoS* and many other important global stress regulators were not affected by *pcnB*-null mutation (Fig S1B) indicating a distinct regulation of stress response gene expression by PAPI. qRT–PCR analysis of 10 select stress response genes from our RNA-Seq data showed stimulated expression on stress exposure and similar up-regulation on *pcnB*-null mutation (Fig 1D). To validate the role of PAPI in stress response, we tested stress sensitivity of WT MG1655, *pcnB*-null mutant (MG-*pcnB*), and ectopically expressed N-terminally FLAG-tagged PAPI (pFLAG$^B$-*pcnB* plasmid transformed in MG1655) under five different stress conditions (osmotic stress, DNA damage/alkylation, acid shock, heat shock, and cold shock). Although MG1655 cells were viable until $10^{-6}$ dilutions in control condition, their growth was compromised under different stresses, which was substantially ameliorated by *pcnB*-null mutation (Fig 1E). Conversely, ectopic PAPI expression increased the sensitivity of MG1655 cells to multiple stresses (Fig 1E). Similarly, the number of viable colonies (CFU/ml) was reduced in WT cells on stress exposure (Fig S1C). This was further accentuated by ectopic FLAG-PAPI expression in different stress conditions (Fig S1C). There was a marked increase in viable colonies on *pcnB*-null mutation under the same stresses. In the growth curve analysis in the presence of stress (NaCl) treatment, WT cells barely reached $OD_{600}$ of ~1.5 on stress treatment in 10 h against ~3.5 $OD_{600}$ under control condition (Fig 1F). *pcnB*-null mutation resulted in a considerable recovery of the growth defect (>2.5 $OD_{600}$) under stress, whereas pFLAG$^B$-*pcnB* transformation rendered cells supersensitive (<0.5 $OD_{600}$ in 10 h) to the same stresses confirming the role of PAPI in stress response.

## Loss of PAPI activity on stress exposure stabilises the expression of stress response mRNAs

Furthermore, we measured half-lives ($T_{1/2}$) of select mRNAs in the presence and absence of stress treatment. We observed >threefold increase in half-lives of mRNAs (*osmY* and *otsA*) on stress exposure and on *pcnB*-null mutation consistent with induced mRNA expression (Fig 2A). Interestingly, there was no further increase in the $T_{1/2}$ of mRNAs in *pcnB*-null mutant cells on stress treatment (Fig 2A) indicating a distinct PAPI-mediated mRNA stabilisation under

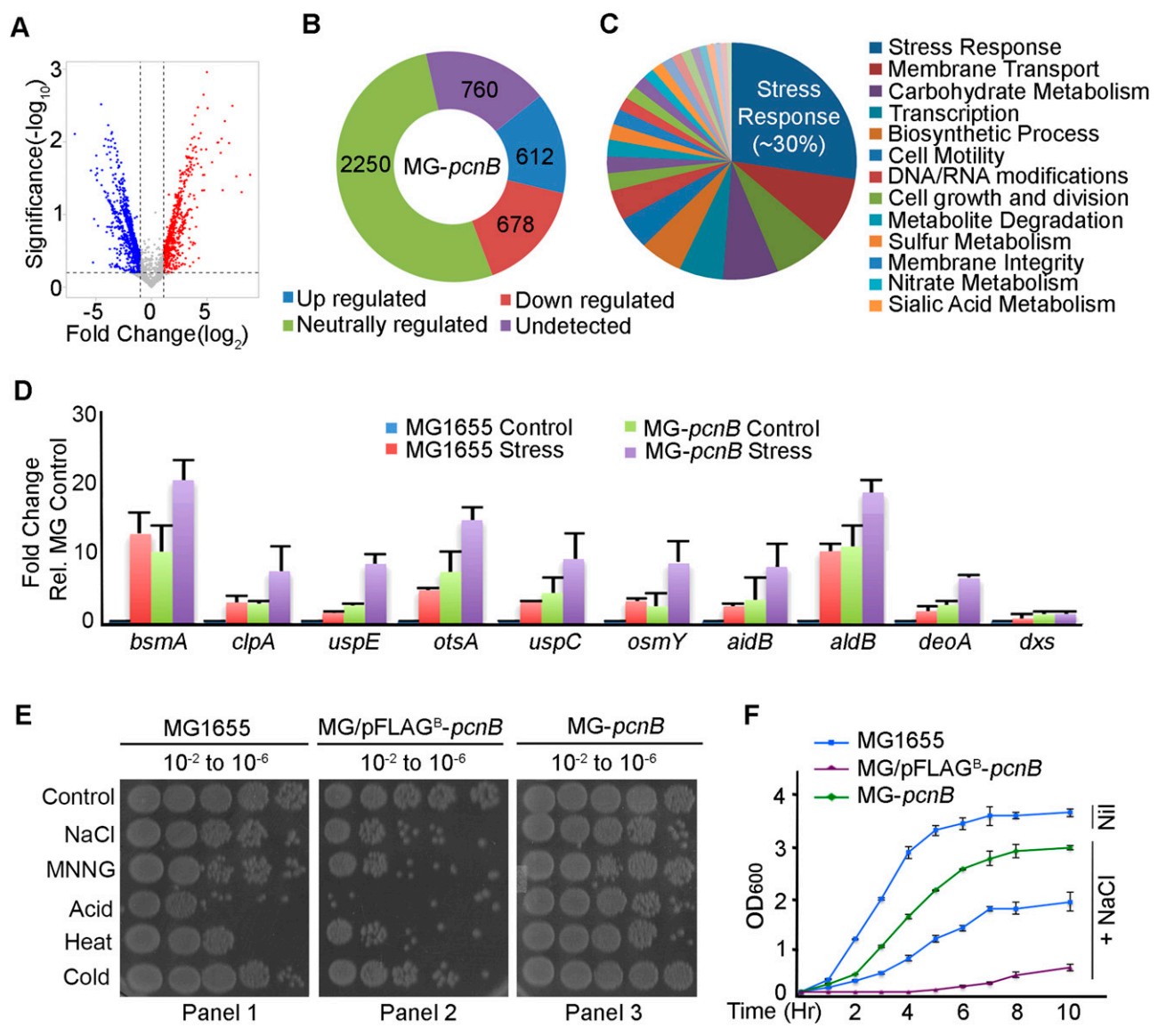

**Figure 1. Genome-wide-RNA-Seq after *pcnB*-null mutation reveals stabilisation of stress response mRNAs.**
**(A)** Volcano plot showing up-regulated and down-regulated genes on *pcnB*-null mutation (MG1655 Δ*pcnB759::kan*, MG-*pcnB*) relative to WT MG1655 cells with threshold values less than −twofold for down-regulation (denoted in blue) and more than +twofold for up-regulation (denoted in red), respectively, for differential gene expression. **(B)** Doughnut plot showing a number of genes in *E. coli* that are altered (down-regulated, up-regulated, and neutrally regulated) from the total genes in *E. coli* (Blattner et al, 1997) in MG-*pcnB* strain compared with MG1655 cells as in (A). Number of undetected genes in our RNA-Seq is also indicated. A complete list of altered genes along with the fold changes is shown in Table S1. **(C)** Functional pathway analysis of up-regulated genes on *pcnB*-null mutation from (B) that reveals multiple cellular functions with most of the genes involved in stress response. **(D)** qRT–PCR analysis of various stress response mRNAs from total RNA isolated from MG1655 and MG-*pcnB* cells in the presence and absence of treatment with different stresses (*bsmA*, *clpA*, and *uspE* were after treatment with $H_2O_2$; *otsA*, *uspC*, and *osmY* were after treatment with NaCl; *aidB*, *aldB*, and *deoA* were after treatment with MNNG; and control *dxs* was after treatment with NaCl). Error bar represents the SEM of n = 3 independent experiments ($P <$ 0.001 for *bsmA*, <0.005 for *deoA*, <0.001 for *clpA*, <0.001 for *otsA*, <0.02 for *osmY*, <0.001 for *uspC*, <0.02 for *uspE*, <0.01 for *aidB*, and <0.001 for *dxs*). **(E)** Dilution spotting of MG1655, *pcnB*-null mutant (MG-*pcnB*), and MG1655 transformed with pFLAG^B-*pcnB* (MG/pFLAG^B-*pcnB*) that ectopically expresses FLAG epitope–tagged PAPI after treatment with different stresses at various dilutions from $10^{-2}$ to $10^{-6}$ on an LB agar plate as indicated. **(F)** Growth curve analysis of strains as in (E) in the presence of stress (NaCl) treatment or untreated control (Nil) as indicated. Error bar represents the SEM of n = 3 independent experiments.

stress. To further understand the PAPI-mediated stabilisation during stress, we assessed in vivo polyadenylation by 3′-RACE assay of a PAPI target *osmY* mRNA (Figs 2B and S2A). We observed a surprising loss of 3′-RACE product on stress treatment similar to *pcnB*-null mutation (Fig 2B) suggesting a reduction in cellular polyadenylation under stress. Control RT–PCR of *dxs* mRNA was not

affected by any of the conditions (Fig 2C). However, there was no reduction in the *pcnB* expression level in the presence of different stress conditions (acid, heat, cold, osmotic, oxidative, and DNA damage) (Fig S2B). Instead, a marginal increase in the *pcnB* mRNA expression was visible (Fig S2B) as opposed to a decrease in the polyadenylation of target mRNAs on stress treatment. We then

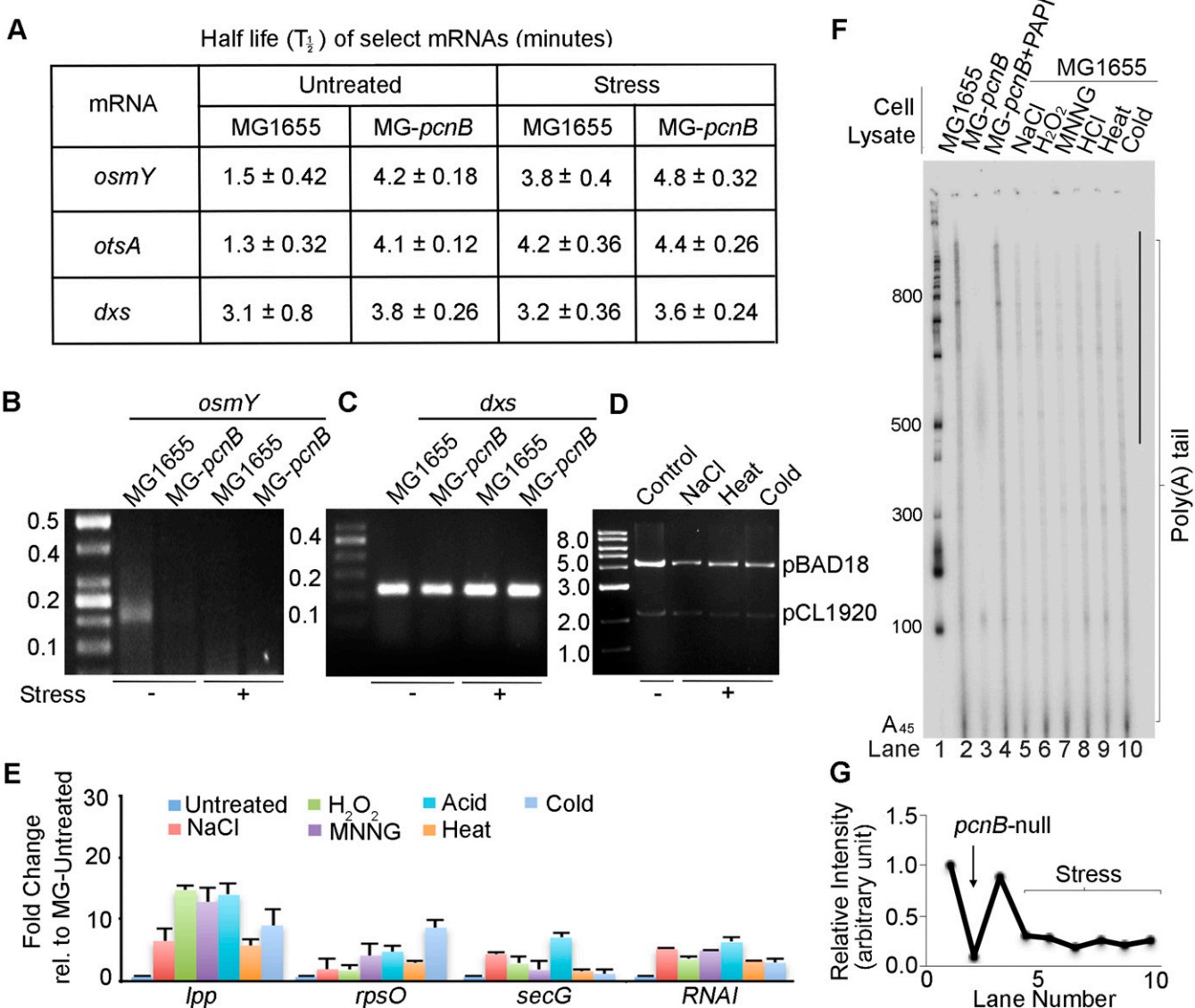

**Figure 2. Stress exposure reduces PAPI polyadenylation activity stabilising target stress response mRNAs.**
**(A)** Half-life ($T_{1/2}$) measurement of stress response mRNAs (*osmY* and *otsA*) and a non-polyadenylated control transcript, *dxs*, after inhibition of transcription with rifampicin from WT MG1655 and MG-*pcnB* mutant strains as indicated. $T_{1/2}$ is expressed in minutes. Data are mean ± SEM of n = 3 independent experiments. **(B)** 3′-RACE assay of stress response–polyadenylated mRNA *osmY* using an engineered oligo-dT primer having a unique sequence at the 3′-end and an *osmY* gene–specific forward primer from total mRNA isolated from MG1655 and MG-*pcnB* strains under conditions as indicated. (Schematic of 3′-RACE assay is shown in Fig S2A.) **(C)** Control RT–PCR of *dxs* mRNA using a pair of primer from the *dxs* cds region. **(D)** Analysis of plasmid content of colE1-based pBAD18 plasmid and control *RNAI*-independent plasmid (pCL1920) isolated from MG1655 cell cultured in the presence of different stresses as indicated. Plasmids were linearised with NdeI before analysis on the gel. **(E)** qRT–PCR analysis of various PAPI target non–stress-related genes from the total RNA isolated from MG1655 cells cultured in the presence of treatment with different stresses as indicated. Fold change mRNA levels were expressed relative to RNA samples from untreated MG1655 cells. Error bar represents the SEM of n = 3 independent experiments ($P < 0.03$ for *lpp*, <0.01 for *rpsO*, <0.002 for *secG*, and <0.02 for *RNAI*). **(F)** In vitro polyadenylation assay carried out on a universal 45-mer RNA oligonucleotide $(UAGGGA)_5A_{15}$ $(A_{45})$ template (Mellman et al, 2008) with active cell lysates prepared from MG1655 cells after treatment with various stressors (lanes 5–10) as indicated. Assay with control lysates from stress-untreated cells (lane 2) and *pcnB*-null mutant (MG-*pcnB*) cells (lane 3), and lysates from MG-*pcnB* supplemented with recombinant His-PAPI (50 nM) (lane 4) is indicated. **(G)** Densitometric quantification of intensities of bands in phosphor images of polyadenylation assays in (F) in arbitrary units expressed as relative intensity with respect to the intensity of MG1655-untreated (control) cell lysates (quantified regions on the gel are indicated). Data are average of n = 3 independent experiments.

assayed PAPI polyadenylation activity from MG1655 cell lysate prepared after treatment with different stressors (NaCl, $H_2O_2$, MNNG, acid, heat, or cold) on a universal polyadenylation template (Mellman et al, 2008). Control polyadenylation assays using mammalian and bacterial PAPI are shown in Fig S2C. Untreated

MG1655 cell lysate exhibited a robust polyadenylation activity that was diminished on *pcnB*-null mutation but reversible by recombinant His-PAPI supplementation (Fig 2F). Strikingly, there was ~60–70% reduction in the PAPI activity on treatment with different stresses (Fig 2F and G). Control assays with oligo-dT

annealing and RNase H digestion confirm polyadenylation by cell lysates on the polyadenylation template (Fig S2E and G). In parallel, recombinant His-PAPI purified from the cell after treatment of the overexpressed cells with different stresses also showed similar reduction in polyadenylation activity compared with His-PAPI purified from stress-untreated cells (Fig S2D and F). Together, these results indicate that there is a reduction in the PAPI activity consistent with induced levels of target mRNAs on stress exposure.

To confirm a decrease in the PAPI activity downstream of stress exposure, we assessed mRNA levels of stress-independent PAPI target mRNAs (rpsO, secG, lpp, and RNAI) in WT MG1655 cells. Concomitant to a decrease in PAP activity, we observed increased endogenous levels of these mRNAs under different stress conditions (NaCl, MNNG, H$_2$O$_2$, heat, cold, and acid) (Fig 2E). Akin to increased levels of RNAI, we also observed a subsequent reduction in the plasmid copy number under different stress conditions (Fig 2D and E). Furthermore, NaCl treatment stimulated the expression of other genes involved in response to different stresses than osmotic stress (bsmA and clpA of oxidative stress response, cspD, uspA, and rmf of cold stress response, and dsdA and aldB of DNA alkylation stress response) irrespective of the stress given (Fig S2H) confirming reduction in PAPI activity as a general stress response mechanism.

### PAPI is phosphorylated at the tyrosine residue that inhibits PAPI polyadenylation activity to regulate stress response gene expression

Earlier, a B. subtilis serine kinase PkrC was shown to phosphorylate PAPI that inhibits PAPI activity in vitro (Jasiecki & Węgrzyn, 2006a). To understand how PAPI activity was reduced under stress, we assessed PAPI phosphorylation both in vivo and in vitro. In vivo, FLAG-PAPI was immunopurified from MG-pcnB cells transformed with pFLAG$^B$-pcnB plasmid that expresses an N-terminally FLAG-tagged PAPI protein. We then analysed tyrosine and serine phosphorylation by Western blotting using phosphoserine- and phosphotyrosine-specific antibodies. To our surprise, we detected tyrosine but not serine phosphorylation on PAPI (Fig 3A). This was then tested using purified recombinant His-PAPI protein from BL21(DE3) that also showed similar tyrosine phosphorylation on PAPI (Fig S3A). To confirm the PAPI tyrosine phosphorylation, we carried out direct phosphoamino acid analysis on $^{32}$P-radiolabelled PAPI after in vitro phosphorylation with stress-treated cell lysates (osmotic and acid) (Tavare et al, 1991; Islas-Flores et al, 1998). Acid-hydrolysed $^{32}$P-radiolabelled PAPI was then analysed in a one-directional TLC. The migration of the radiolabelled amino acid residue observed on the TLC after the hydrolysis was equivalent to that of phosphotyrosine standard but not phosphoserine standard (Fig 3B), revealing that PAPI is phosphorylated at the tyrosine residue by stress cell lysates.

Then, we assessed PAPI tyrosine phosphorylation by Western blot analysis of FLAG-PAPI immunopurified from MG-pcnB cells transformed with pFLAG$^B$-pcnB plasmid after treatment with different stresses. We observed a striking increase in the tyrosine phosphorylation on PAPI after treatment with different stresses (NaCl, H$_2$O$_2$, MNNG, acid, heat, and cold) (Fig 3C). There was >5–10-fold increase in the phosphorylation under different stresses

irrespective of the types of stress given (Figs 3C and S3B). A similar increase in the PAPI tyrosine phosphorylation was also observed from the Western blot analysis of recombinant His-PAPI purified after treatment of the cell with different stresses (Fig S3C and D). This was further tested in an in vitro kinase assay using recombinant His-PAPI and radiolabelled $^{32}$P-γATP with active MG1655 cell lysates prepared after treatment with different stressors. Consistently, there was induction of kinase activity on His-PAPI protein with cell lysates prepared after treatment with different stresses (NaCl, H$_2$O$_2$, MNNG, acid, heat, and cold) (Fig 3D). Kinase activity on His-PAPI from control untreated cell lysate was negligible in the reaction. This was further validated in the direct phosphoamino acid analysis of in vitro–phosphorylated PAPI in a one-directional TLC experiment (Fig 3E). We consistently detected increased phosphorylated tyrosine from the in vitro–phosphorylated PAPI using stress-treated cell lysate (NaCl) compared with the control untreated cell lysates (Fig 3E). Western blot analysis using phosphotyrosine-specific antibody after in vitro phosphorylation also detected significant tyrosine phosphorylation on His-PAPI with stress-treated cell lysates (NaCl and MNNG) (Fig 3G). There was no significant phosphorylation on His-PAPI by non–stress-treated cell lysate. Together, these results demonstrate that PAPI is phosphorylated at the tyrosine residue that is stimulated during stress response.

Furthermore, we assessed the effect of PAPI phosphorylation on its polyadenylation activity by an in vitro polyadenylation assay of His-PAPI after in vitro phosphorylation with stress-primed and unprimed cell lysates (Fig 3H). There was a dramatic reduction in the PAPI polyadenylation activity after phosphorylation with stress (NaCl)-primed MG1655 cell lysates (Figs 3H and S3E and F). Although there was >5–10-fold decrease in PAPI activity post-phosphorylation with stress-primed lysates, PAPI treated with unprimed cell lysates did not show a significant effect on the PAPI activity (Figs 3H and S3E and F). To further confirm the effect of PAPI phosphorylation on its activity, we immunopurified FLAG-PAPI from cells after NaCl treatment. It was then dephosphorylated with λ-phosphatase in the presence and absence of a phosphatase inhibitor (sodium vanadate), and in vitro polyadenylation assay was carried out (Figs 3I and S3H). We observed stimulated PAPI activity after dephosphorylation of FLAG-PAPI by λ-phosphatase (Figs 3I and S3H). However, the increased PAPI polyadenylation activity on λ-phosphatase treatment was ameliorated in the presence of sodium vanadate. We then used a specific bacterial tyrosine phosphatase (YopH protein from Yersinia pestis) to dephosphorylate PAPI in the presence and absence of sodium vanadate before the in vitro polyadenylation assay (Figs 3I and S3H). Consistently, we observed an increase in PAPI activity on YopH treatment that was lost in the presence of sodium vanadate (Figs 3I and S3H). Western blot showing a decrease in the phospho-PAPI level on inhibition with λ-phosphatase or YopH and its inhibition by sodium vanadate is shown in Fig S3G. Together, these results show that stress-induced tyrosine phosphorylation inhibits PAPI polyadenylation activity.

### Identification of stress-induced phosphorylated tyrosine residue on PAPI

To identify the stress-induced tyrosine phosphorylation site(s) on PAPI, we first employed in silico prediction software (NetPhos 3.1) to

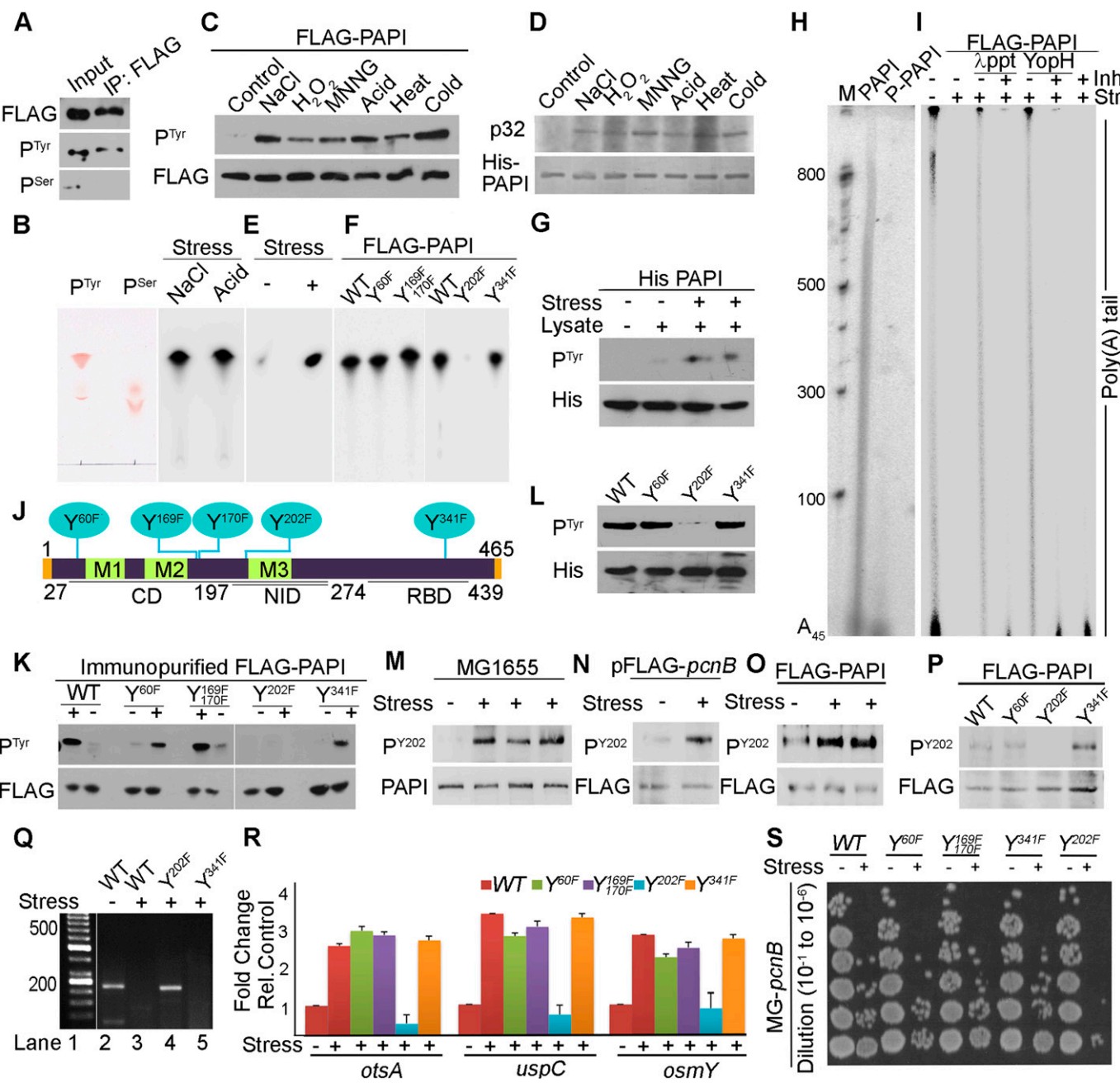

**Figure 3. Induced PAPI phosphorylation at tyrosine 202 residue regulates PAPI polyadenylation activity under stress.**

**(A)** Western blot analysis using phosphotyrosine (P$^{Tyr}$), phosphoserine (P$^{Ser}$), and control FLAG epitope tag (FLAG)–specific antibodies from immunopurified FLAG-PAPI from MG-*pcnB* cells after transformation with pFLAG$^B$-*pcnB* plasmid. **(B)** Direct phosphoamino acid analysis using one-directional TLC after acid hydrolysis of in vitro–phosphorylated recombinant His-PAPI in the presence of γ$^{32}$P-ATP using osmotic (NaCl) and acid (HCl)-treated cell lysates. Ninhydrin staining of standard cold phosphoserine and phosphotyrosine amino acids on the TLC plate is indicated. **(C)** Western blot analysis using phosphotyrosine (P$^{Tyr}$) and FLAG antibody of FLAG-PAPI immunopurified from MG-*pcnB* cells transformed with pFLAG$^B$-*pcnB* plasmid in the presence and absence of treatment with different stresses as indicated. Quantification of the relative intensities of each band of the blot is shown in Fig S3B. **(D)** In vitro kinase assay of recombinant His-PAPI with MG1655 cell lysates prepared after treatment with various stressors as indicated. Coomassie-stained gel of control-purified His-PAPI protein is indicated below. **(E)** Direct phosphoamino acid analysis as in (B) but using stress (NaCl)-untreated or stress-treated cell lysates as indicated. **(F)** Direct phosphoamino acid analysis as in (B) of WT and different phosphomutants of PAPI (WT, Y60F, Y169F, Y170F, Y202F, and Y341F, respectively) after in vitro phosphorylation using stress-treated cell lysates as indicated. **(G)** Western blot analysis using phosphotyrosine-specific antibody of recombinant His-PAPI after cold in vitro phosphorylation with MG1655 cell lysates prepared with and without stress (NaCl and HCl) treatment. **(H)** In vitro polyadenylation assay of recombinant His-PAPI protein after in vitro phosphorylation with stress (NaCl)-treated cell lysate. Phosphorylated His-PAPI after the reaction is as in (G). **(I)** In vitro polyadenylation assay of PAPI protein after in vitro phosphorylation with untreated or stress (NaCl)-treated cell lysate. Dephosphorylation with either λ-phosphatase (λppt) or bacterial-specific tyrosine-protein phosphatase (YopH) in the presence or absence of phosphatase inhibitor, sodium vanadate (Inh), as indicated. Control polyadenylation assays with sodium vanadate without phosphatase treatment are also shown. Changes in the phosphorylation status of PAPI in these reactions are shown in Fig S3G. **(J)** Schematic of PAPI protein domain organisation showing putative phosphorylation sites predicted in silico using the NetPhos 3.1 software from the PAPI primary sequence (Blom et al, 1999). Catalytic Domain (CD) with three different

shortlist putative tyrosine phosphorylation sites on PAPI. We detected a number of putative sites of which five distinct sites (four in the catalytic domain [tyrosine residues Y60, Y169, Y170, and Y202], and one at the C-terminal RNA-binding domain [tyrosine 341, Y341]) were shortlisted as they were above the threshold as determined by the software (Figs 3J and S3I). To identify the phosphorylated tyrosine residue under stress, we mutated each putative tyrosine (Y) residue to phenylalanine (F) by site-directed mutagenesis (phosphodeficient mutations) (Y60F, Y169F, Y170F, Y202F, and Y341F) (Fig 3J) (Smith et al, 1994). These mutations were created on pFLAG$^B$-*pcnB* and pET-*pcnB* plasmid constructs. First, we transformed each of the mutant constructs (pFLAG$^B$-*pcnB-Y60F*, pFLAG$^B$-*pcnB-Y169F*, *Y170F*, pFLAG$^B$-*pcnB-Y202F*, and pFLAG$^B$-*pcnB-Y341F*) and control WT *pcnB* construct (pFLAG$^B$-*pcnB*) in MG-*pcnB* cells. WT and mutant FLAG-PAPI proteins were immunopurified after treatment with NaCl stress and analysed for tyrosine phosphorylation by Western blot analysis using phosphotyrosine antibody. Strikingly, Y202F mutation at the nucleobase-interacting pocket but not other mutations (Y60F, Y169F, Y170F, and Y341F) abolished induced PAPI tyrosine phosphorylation on stress treatment (Fig 3K). Similarly, in an in vitro kinase assay with stress-treated cell lysate, we did not detect phosphotyrosine from Y202F mutant PAPI (Fig 3L). Consistently, using the phosphoamino acid analysis, we showed that phosphotyrosine was not detected from in vitro–phosphorylated Y202F mutant PAPI, whereas it was detected from the WT and other control phosphomutants (Y60F, Y169F, Y170F, and Y341F) employed (Fig 3F).

To further characterise the Y202 phosphorylation on PAPI, an antibody was raised against a Y202-phosphorylated PAPI peptide (IRLIGNPETRY[p]REDPVRMLR). The purified antibody detected the endogenous phosphorylated PAPI (referred to as Y202-phospho-PAPI [PAPI-P$^{Y202}$] and the antibody as Y202-phosphoantibody [P$^{Y202}$]) in different stress-treated and untreated MG1655 cells (Figs 3M and S3J). Competition with both phosphopeptide and non-phosphopeptide demonstrated the specificity of the Y202-phosphoantibody. The Y202-phosphopeptide specifically competed for with the Y202-phosphoantibody but not with the same non-phospho-PAPI peptide (Fig S3J). Moreover, our Y202-phosphoantibody was further tested in an experiment using ectopic expressions of FLAG-PAPI in MG1655 (Fig 3N). Consistently, Y202-phosphoantibody also detected tyrosine-phosphorylated FLAG-PAPI in immunopurified FLAG-PAPI from the ectopically expressed cells (Fig 3O). The phosphorylation

was induced on stress treatment (NaCl and HCl) as in the case of endogenous phosphorylation (Fig 3N and O). Consistently, Western blot analysis with Y202-phosphoantibody also showed tyrosine phosphorylation on WT PAPI that was lost on Y202F mutation but not on other phosphomutations tested (Y60F and Y341F) (Fig 3P). Together, these results demonstrate that PAPI is phosphorylated at the Tyr-202 residue that is induced under stress.

### Tyrosine-202 phosphorylation at the PAPI catalytic domain inhibits PAPI polyadenylation activity and induces stress response gene expression

To further understand the effect of Y202 phosphorylation on PAPI activity, we carried out an in vitro polyadenylation assay of WT and Y202F mutant FLAG-PAPI immunopurified in the presence and absence of stress (NaCl) treatment (Fig S4A and B). For this purpose, MG-*pcnB* cells transformed with WT *pcnB* construct (pFLAG$^B$-*pcnB*) or mutant pFLAG$^B$-*pcnB-Y202F* or pFLAG$^B$-*pcnB-Y341F* constructs were used for FLAG-PAPI purification. Although the WT FLAG-PAPI activity was diminished after stress treatment, polyadenylation activity of Y202F FLAG-PAPI was unaffected by stress treatment (Fig S4A and B). The activity of control Y341F FLAG-PAPI was reduced similar to WT PAPI. Accordingly, in a 3′-RACE assay, cellular polyadenylation on *osmY* mRNA was compromised on stress treatment that was augmented by PAPI Y202F mutation but not by other control mutations (Fig 3Q), indicating that Y202 phosphorylation regulates PAPI activity. To further confirm this, we measured mRNA levels of PAPI target stress response genes (*uspC*, *otsA*, and *osmY*) from MG-*pcnB* cells transformed with WT and respective mutant *pcnB* constructs of pFLAG$^B$-*pcnB*. Although all the three mRNAs were stimulated on stress treatment in the WT PAPI-expressing cells, mutation of Y202F failed to induce stress response gene expression (Fig 3R). Consequently, in cellular stress tolerance assessment, although the growth of pFLAG$^B$-*pcnB*–transformed cells was reduced on stress treatment, introduction of *pcnB-Y202F* mutation but no other mutations (*Y60F*, *Y169F*, *Y170F*, and *Y341F*) rendered cells supersensitive to stress in both dilution plating and viable colony counting (Figs 3S and S4C). There was ~twofold to fourfold reduction in viable colonies on stress treatment that was further accentuated (~5–10-fold) specifically by *pcnB-Y202F* mutation (Fig S4C). There was marginal cellular growth with *pcnB-Y202F* mutation compared with WT under stress treatment (Fig S4C).

motiffs (M1, M2 and M3), Nucleotide Interaction Domain or pocket (NID), and RNA Binding Domain (RBD) are shown. Five putative tyrosine phosphorylation sites and respective mutations introduced are indicated (Y60F, Y169F, Y170F, Y202F, and Y341F). **(K)** Western blot analysis of FLAG-PAPI using phosphotyrosine-specific antibody and control anti-FLAG epitope tag antibody immunopurified from MG-*pcnB* cells transformed with pFLAG$^B$-*pcnB* plasmid harbouring respective mutations in the presence and absence of stress (NaCl) treatment as indicated. **(L)** Western blot analysis of recombinant His-PAPI WT and phosphomutants as indicated after in vitro phosphorylation with stress-primed MG1655 cell lysates using phosphotyrosine-specific antibody. Control Western blot using His-Tag–specific antibody is shown below. **(M)** Western blot analysis of PAPI endogenous phosphorylation using phosphotyrosine Y202-specific antibody (P$^{Y202}$) from MG1655 cell lysates after treatment with different stresses (osmotic stress, acid stress, and cold shock). **(N)** Western blot analysis using phosphotyrosine Y202-specific antibody (P$^{Y202}$) from exogenously expressed FLAG-PAPI in MG-*pcnB* cells after treatment with stress. Control Western blot using FLAG epitope tag–specific antibody is shown below. **(O)** Western blot analysis using phosphotyrosine Y202–specific antibody (P$^{Y202}$) for immunopurified FLAG-PAPI from MG1655 cells transformed with pFLAG$^B$-*pcnB* constructs and treated with different stress or untreated cells as indicated. Control Western blot using FLAG epitope tag–specific antibody is shown below. **(P)** Western blot analysis of FLAG-PAPI using phosphotyrosine Y202-specific antibody (P$^{Y202}$) and control FLAG epitope tag antibody from MG-*pcnB* cells transformed with pFLAG$^B$-*pcnB* plasmid harbouring different phosphomutations (Y60F, Y202F, and Y341F) as indicated in the presence of stress (NaCl) treatment. **(Q)** 3′-RACE assay of *osmY* mRNA from total mRNA isolated from MG-*pcnB* cells transformed with WT pFLAG$^B$-*pcnB* plasmid (WT) and mutant constructs pFLAG$^B$-*pcnB-Y202F* and pFLAG$^B$-*pcnB-Y341F* in the presence and absence of stress (NaCl) treatment as indicated. **(R)** qRT–PCR analysis of stress response genes (*otsA*, *uspC*, and *osmY*) from total mRNA isolated from MG-*pcnB* cells containing WT pFLAG$^B$-*pcnB* plasmid (WT) and mutant constructs pFLAG$^B$-*pcnB-Y60F*, pFLAG$^B$-*pcnB-Y169F, 170F*, pFLAG$^B$-*pcnB-Y202F*, and pFLAG$^B$-*pcnB-Y341F* in the presence of stress (NaCl) treatment. Error bar represents the SEM of n = 3 independent experiments ($P < 0.002$ for *otsA*, <0.001 for *uspC*, and <0.02 for *osmY*). **(S)** Dilution spotting of stress-treated and untreated cultures of the same strains as in (R) in the presence and absence of NaCl stress treatment.

Similarly, WT pFLAG[B]-*pcnB*–expressing cells grew ~1.2 $OD_{600}$ on stress treatment against ~0.5 $OD_{600}$ of *pcnB-Y202F* mutant on stress exposure (Fig S4D). There was no marked difference between the WT and other control *pcnB* mutants (*Y60F*, *Y169F*, *Y170F*, and *Y341F*) in the growth curve analysis (Fig S4D).

To understand how Y202 phosphorylation affects PAPI poly-adenylation activity, we analysed the Y202 phosphorylation–induced structural changes using MD simulations spanning 300 ns for PAPI and Y202-phospho-PAPI. In simulations initiated from PAPI, the root mean square deviation of protein backbone of PAPI was significantly increased after Y202 phosphorylation throughout the 300-ns simulation (Fig 4A, top panel). This suggests that PAPI undergoes an overall structural alteration on Y202 phosphorylation and that it is less stable than the WT protein. Moreover, the Y202 phosphorylation perturbed several interactions including hydrogen bonds formed in the WT protein. We observed a new hydrogen bond established between Y202 residue and threonine 227 (T227) on phosphorylation with a bond length of ~2.7 Å (Fig 4A, lower panel, and Fig S4E) that remained intact throughout the simulation time (Fig 4A, lower panel). The same bond distance was 5.1 Å in the case of non-phosphorylated PAPI and is insufficient to establish the H-bond. This hydrogen bond formation could affect the ATP-binding pocket by repositioning of amino acids in the pocket. Therefore, we compared simulations of ATP-bound form of both Y202-phospho- and non-phospho-PAPI. Strikingly, there was an overall reduction in the number of hydrogen bonds formed in the ATP-binding pocket in the MD-simulated structure of Y202-phosphorylated PAPI compared with the WT PAPI (Fig S4F). We observed a loss of six of seven hydrogen bonds that were known to stabilise the ATP-PAPI interaction after Y202 phosphorylation (Toh et al, 2011) (Fig 4B). These include direct ATP interactions (residues R208, R214, and R161) and intramolecular interactions within the nucleobase-interacting pocket that stabilises ATP binding (D162 and R211, D205 and R211, and G204 and R208). The bond distance of ATP direct interactions was respectively altered from 2.7 to 14.2 Å for R208, 3.6 to 15 Å for R161, and 3.7 to 8 Å for R214, thus losing a stable interaction with ATP on Y202 phosphorylation (Fig 4B). Among the other critical intramolecular interactions, the distance between D205 and R211 (that are known to affect PAPI activity) was increased from 3.7 to 8.3 Å in addition to the loss of polar contacts between the two. As a result, as simulation progresses, ATP leaves the binding pocket from the phospho-Y202 PAPI unlike in the WT PAPI (Figs 4A and B and S4F). Together, these results suggest that Y202 phosphorylation could induce allosteric changes in the ATP-binding pocket of PAPI that can lead to reduction in its polyadenylation activity.

### Wzc tyrosine kinase phosphorylates PAPI at Y202 to regulate PAPI activity under stress

To identify the tyrosine kinase responsible for PAPI-Y202 phosphorylation, we tested phosphorylation of PAPI using mutants of earlier known bacterial tyrosine kinases Δ*wzc-758*::*kan* (MG-*wzc*), Δ*etk-725*::*kan* (MG-*etk*), and Δ*bipA733*::*kan* (MG-*bipA*) and Δ*ydiB-766*:: *kan* (MG-*ydiB*) under stress (Grangeasse et al, 2003, 2007, 2012) using phosphotyrosine antibody. We observed a loss of induced PAPI tyrosine phosphorylation under stress specifically in *wzc*-null

mutant but not in other mutants (Fig 4C), suggesting that Wzc is the kinase that phosphorylates PAPI. We then validated these results using Y202-phospho-PAPI–specific antibody, and we consistently observed no endogenous PAPI tyrosine phosphorylation detected in the presence of *wzc*-null mutation (Fig 4D). However, it was detected in WT and in the presence of other kinase mutants (Fig 4D). Furthermore, we carried out an in vitro phosphorylation assay with His-PAPI and stress-primed cell lysates from various kinase mutants. MG-*wzc* cell lysate but not other mutant cell lysates failed to phosphorylate His-PAPI when tested by Western blot analysis with Y202-phospho-PAPI–specific antibody (Fig 4E). Similarly, there was no incorporation of radiolabelled [32]P (from [32]P-ATP) in an in vitro kinase assay with cell lysates from MG-*wzc*, whereas WT MG1655 and other mutant cell lysates showed kinase activity towards His-PAPI (Fig S5A). To confirm the phosphorylation of PAPI by Wzc kinase, we employed a kinase-enfeebled mutation, K540R (that abrogates its autophosphorylation and target Ugd phosphorylation) (Grangeasse et al, 2002, 2003), or WT ectopically expressed in MG-*wzc* cells and analysed for Y202-phospho-PAPI (Fig 4F). We showed a loss of Y202 phosphorylation in MG-*wzc* mutant cells that was rescued by WT FLAG-Wzc expression but not by the WzcK540R (Fig 4F). Finally, we immunopurified FLAG-Wzc and carried out in vitro phosphorylation of recombinant WT PAPI and various phosphomutants of PAPI (Y202F, Y60F, Y169F, Y170F, and Y341F) in vitro. Strikingly, although purified FLAG-Wzc exhibited kinase activity towards WT and other PAPI mutants (Y60F, Y169F, Y170F, and Y341F), it failed to exhibit kinase activity towards the Y202F PAPI mutant (Figs 4G and S5B). Consistently, in phosphoamino acid analysis, we detected phosphotyrosine residue from in vitro–phosphorylated PAPI using purified FLAG-Wzc protein but not from kinase-enfeebled WzcK540R protein (Fig 4H). Together, these results confirm that Wzc is the stress-responsive tyrosine kinase that phosphorylates PAPI at the Y202 residue.

Furthermore, to verify whether the reduction in polyadenylation activity was due to phosphorylation of PAPI by Wzc kinase, we performed an in vitro polyadenylation assay using cell lysates from both WT and MG-*wzc*–null mutants after stress treatment. Consistent with previous findings, we observed a reduction in the polyadenylation activity of PAPI after treatment with WT cell lysate that was rescued by *wzc*-null mutation (Fig 4I). We then carried out an in vitro polyadenylation after phosphorylation of PAPI with purified Wzc. Consistently, we observed reduced PAPI activity after phosphorylation with purified FLAG-Wzc protein (Fig 4J). Concomitantly, there was no significant up-regulation of PAPI target mRNA (*bsmA*, *otsA*, and *osmY*) expression on stress treatment in the presence of *wzc*-null mutation (Fig 4K). Subsequently, in the cellular tolerance assay, *wzc*-null mutant strain rendered cells more stress-sensitive similar to PAPI-Y202 mutation (Fig 4L). Although WT cells reached >1.5 $OD_{600}$ in 12 h under stress, *wzc*-null mutant cells reached <0.5 $OD_{600}$ at the same time under stress (Fig 4L). Likewise in the dilution plating, *wzc*-null mutant cells showed heightened sensitivity to stress treatment compared with WT and other mutants (Fig S5C), reiterating that Wzc is the kinase that phosphorylates PAPI under stress. In line with these findings, although *wzc* mutation makes cells stress-sensitive than the WT cells (Fig 4L), there was no effect of *wzc*-null mutation in the presence of *pcnB*-null mutation (Fig 5A), demonstrating that Wzc functions through

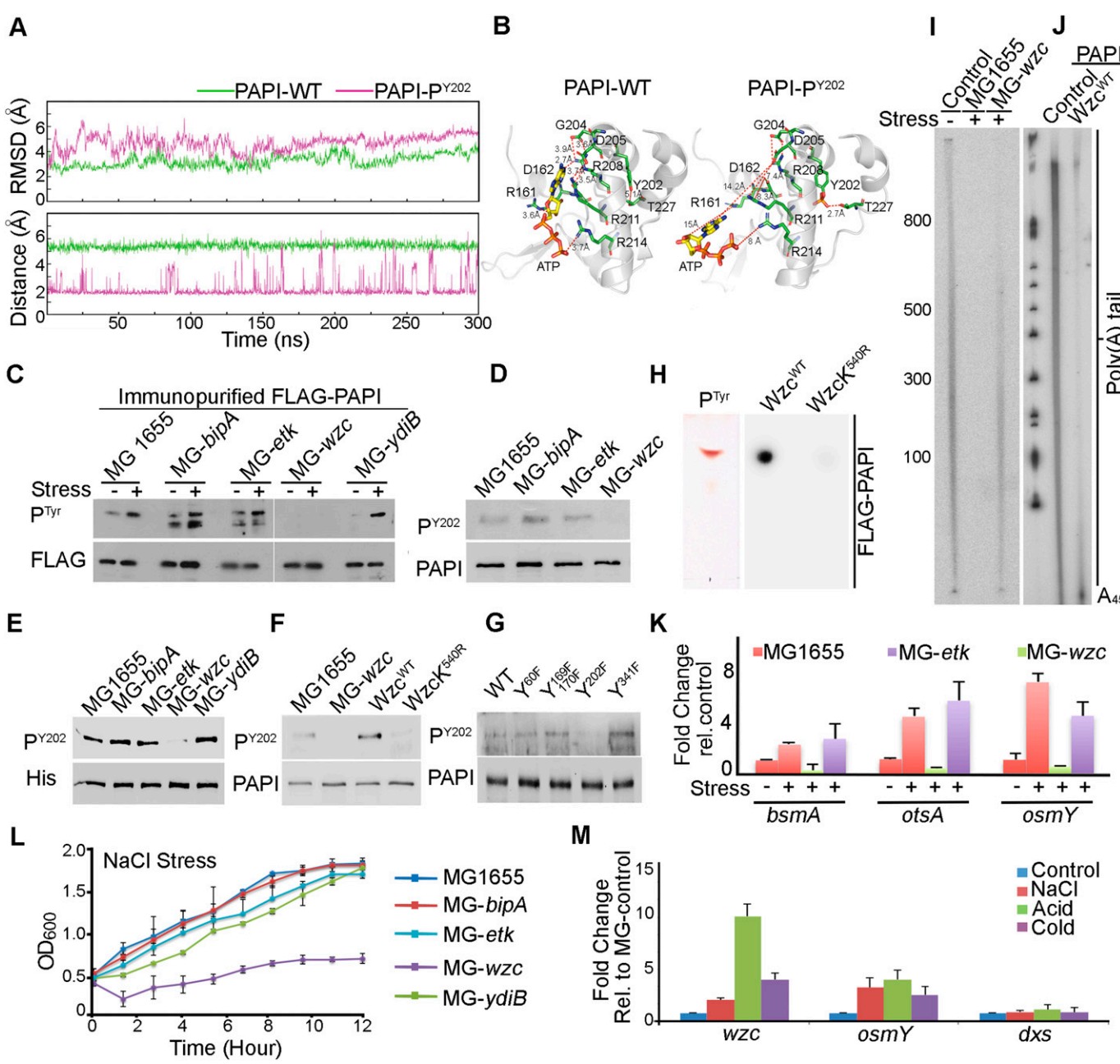

**Figure 4. Wzc tyrosine kinase phosphorylates PAPI at Y202 residue that regulates general stress response in *E. coli*.**
**(A)** Root mean square deviation analysis of apo form of WT PAPI (green) versus Y202-phosphorylated PAPI (PAPI-P^Y202) (magenta) after MD simulations spanning 300 ns (upper panel). Molecular simulations for the interaction between Y202 and T227 in the WT versus Y202-phospho-PAPI spanning 300 ns (lower panel). **(B)** Comparison of ATP recognition by WT and Y202-phospho-PAPI in the regions spanning catalytic and nucleobase interaction domain. Key residues that are involved in the interaction with ATP and the distance of the interactions that are altered on Y202 phosphorylation are indicated. **(C)** Western blot analysis of immunopurified FLAG-PAPI using phosphotyrosine-specific antibody (P^Tyr) and FLAG epitope tag antibody (FLAG) from MG1655, MG-*bipA*, MG-*etk*, MG-*wzc*, and MG-*ydiB* strains in the presence and absence of stress (NaCl) treatment. Control Western blot using FLAG epitope tag–specific antibody is shown below. **(D)** Western blot analysis of endogenous PAPI phosphorylation using phosphotyrosine Y202–specific antibody (P^Y202) in various tyrosine kinase mutant strains as in (C) as indicated. **(E)** Western blot analysis using phosphotyrosine Y202–specific antibody of purified His-PAPI after in vitro phosphorylation with stress-primed cell lysates of various kinase mutant strains as in (C) as indicated. **(F)** Western blot analysis of endogenous phosphorylation of PAPI using phosphotyrosine Y202-specific antibody in WT, *wzc* mutant, *wzc* mutant expressing FLAG-tagged WT Wzc (Wzc^WT), or kinase-enfeebled WzcK540R mutant (WzcK^540R), respectively. Control PAPI blot is shown. **(G)** Western blot analysis of PAPI tyrosine phosphorylation of WT and different PAPI phosphomutations (Y60F, Y169F, Y170F, Y202F, and Y341F) after in vitro phosphorylation with purified FLAG-Wzc protein kinase. **(H)** Direct phosphoamino acid analysis as in Fig 3B of recombinant PAPI protein phosphorylated with WT Wzc kinase (Wzc^WT) and kinase-enfeebled Wzc mutant K540R (WzcK^540R) (Grangeasse et al, 2002, 2003) as indicated. **(I)** In vitro polyadenylation assay of PAPI after phosphorylation with stress-primed cell lysates from WT and *wzc* mutant cells. **(J)** In vitro polyadenylation assay of purified PAPI with and without phosphorylation with purified FLAG-Wzc. **(K)** qRT–PCR analysis PAPI target genes from MG1655, *etk* mutant (MG-*etk*), and *wzc* mutant (MG-*wzc*) cells after treatment with NaCl stress (P < 0.002 for *bsmA*, <0.01 for *otsA*, and <0.02 for *osmY*). **(L)** Growth curve analysis of various kinase mutants as in (E) in the presence of NaCl stress treatment. Error bar represents the SEM of n = 3 independent cultures. **(M)** qRT–PCR analysis of *wzc* transcript in MG1655

PAPI phosphorylation. Moreover, we also observed no additional effect of *wzc*-null mutation in the presence of *pcnB-Y202F* phosphomutation on the bacterial stress sensitivity assay (Fig 5B). Accordingly, there was no effect of *wzc*-null mutation on the expression of stress response mRNAs (*osmY* and *otsA*) in the presence of *pcnB* mutation (Fig 5C). Together, these results reveal that Wzc tyrosine kinase phosphorylates Y202 residue at the PAPI catalytic domain to regulate general stress response in *E. coli*.

To further assess how Wzc is regulated during stress, we analysed both *wzc* expression and Wzc autophosphorylation during stress treatment. Transcription of *wzc* operon is controlled by the sigma factor RpoE ($\sigma$24) in addition to the general sigma factor RpoD (Fig S5D) (Rhodius et al, 2006). Earlier studies also showed induced *wzc* expression during osmotic shock, independent of RpoS (Hagiwara et al, 2003; Ionescu & Belkin, 2009). Consistently, we observed an increased *wzc* transcript level in the presence of different stresses (Fig 4M). The control stress-regulated transcript, *osmY*, was also equally induced under all stresses but not the control non–stress-related transcript *dxs* (Fig 4M). However, we did not observe any significant difference in the Wzc autophosphorylation in the presence and absence of stress treatment (Fig S5E), whereas there was a significant increase in the PAPI phosphorylation. These results indicate that *wzc* expression is regulated under stress that further controls PAPI activity during stress response.

### PAPI phosphorylation–mediated stress response mechanism is distinct and operates downstream of the RpoS-mediated pathway

We have shown that diminished PAPI activity on stress exposure stimulates stress response gene expression. To assess a potential transcriptional effect on the PAPI-mediated stimulation of stress response gene expression, we generated a reporter construct of a stress response gene *osmY* (expressing a FLAG epitope–tagged OsmY) from its own 3′-UTR but driven from a stress-insensitive promoter (pFLAG[B]-*osmY*[NS-P]) (Fig S5F). We confirmed the PA-tail addition at the mRNA 3′-end by sequencing of the UTR region after 3′-RACE assay (Fig S5G). We observed ~fourfold increase in FLAG-OsmY protein expression (detected with anti-FLAG antibody) on stress treatment in WT cells similar to *pcnB*-null mutation (Fig 5D), suggesting a *pcnB*-specific non-transcriptional effect on increased *osmY* expression. There was a consistent enhancement of a reporter *FLAG-osmY* mRNA level (forward primer from the *FLAG* region and reverse primer from the *osmY* cds region) on stress (NaCl) treatment similar to *pcnB*-null mutation (Fig 5E). However, there was no further enhancement of the reporter *FLAG-osmY*/FLAG-OsmY expression on stress exposure after *pcnB*-null mutation in both qRT–PCR and Western blot analyses (Fig 5D and E). This demonstrates that the PAPI-mediated stabilisation of stress response mRNA is distinct and independent of transcriptional induction. However, both reporter mRNA and protein expressions from the control reporter construct with stress-sensitive *osmY* promoter (pFLAG[B]-*osmY*[SS-P]) showed an induced expression of

*FLAG-osmY*/FLAG-OsmY under stress after *pcnB*-null mutation in both qRT–PCR and Western blot analyses (Fig 5F and G). This is consistent with mRNA expressions of other stress response genes that were induced to some extent on stress treatment in the presence of *pcnB* mutation (Fig 1D).

To further analyse how PAPI-mediated stress response mechanism functions in concert with the RpoS-mediated transcriptional pathway (Battesti et al, 2011), we employed *rpoS*-null mutant strain, *ΔrpoS746::kan* (MG-*rpoS*), and constructed *rpoS-pcnB* double mutant (*ΔrpoS746, ΔpcnB759::kan*) in MG1655 cells by removing the kanamycin cassette from MG-*rpoS* using pCP20 and P1 transduction from MG-*pcnB* strain. qRT–PCR analysis of stress response mRNAs (*osmY*, *aidB*, and *uspA*) showed increased mRNA levels on stress treatment in the presence and absence of *pcnB*-null mutation that was compromised on *rpoS* mutation (Fig 5H). Strikingly, *pcnB* mutation has no significant effect on the mRNA expression levels in the *rpoS* mutant background under stress (Fig 5H). Moreover, *wzc*-null mutation did not have any additional effect on the gene expression in the presence of *rpoS*-null mutation (Fig 5H). Concomitantly, the expression of WT *pcnB* or *pcnB-Y202F* mutant did not affect significantly the expression levels of stress response mRNAs after *rpoS* mutation (Fig S5H). Together, these results indicate that *pcnB*-mediated stabilisation of stress response mRNAs is downstream of *rpoS*-mediated transcriptional stimulation during stress. Furthermore, we tested stress tolerance of *pcnB*-null mutation and *wzc*-null mutation in the presence of *rpoS* mutation under stress. As reported earlier, we observed hypersensitivity of *rpoS* mutant strain compared with WT MG1655 cells under stress treatment (NaCl) (Kaasen et al, 1992) in both growth curve analysis and the viable colony counting (Figs 5I and S5I). *pcnB*-null or *wzc*-null mutation did not show any discernible effect on the growth sensitivity in the presence of *rpoS* mutation (Figs 5I and S5I). Together, our results established a distinct PAPI tyrosine phosphorylation–controlled stress response mechanism that operates downstream and in concert with RpoS transcriptional induction of stress response gene expression.

## Discussion

Polyadenylation at the structured 3′-end is a key post-transcriptional modification of mRNAs in bacteria that initiates 3′- to 5′-exonucleolytic degradation, a process critical for mRNA turnover and quality control (Régnier & Arraiano, 2000; Mohanty & Kushner, 2011; Hajnsdorf & Kaberdin, 2018). Emerging studies have established direct roles of PAPI-mediated polyadenylation in functional gene expression in the number of cellular functions including the general stress response (Aiso et al, 2005; Joanny et al, 2007; Reichenbach et al, 2008; Carabetta et al, 2010; Maes et al. 2012, 2013, 2016; Mohanty & Kushner, 2012; Sinha et al, 2018; Mohanty et al, 2020; Francis & Laishram, 2021). We have shown a distinct PAPI activity–controlled stress response mechanism that operates in

cells treated with various stresses as indicated. Control stress response transcript *osmY* and non–stress-related transcript *dxs* are shown. Error bar represents the SEM of n = 3 independent experiments ($P < 0.002$ for *wzc*, <0.01 for *dxs*, and <0.02 for *osmY*).

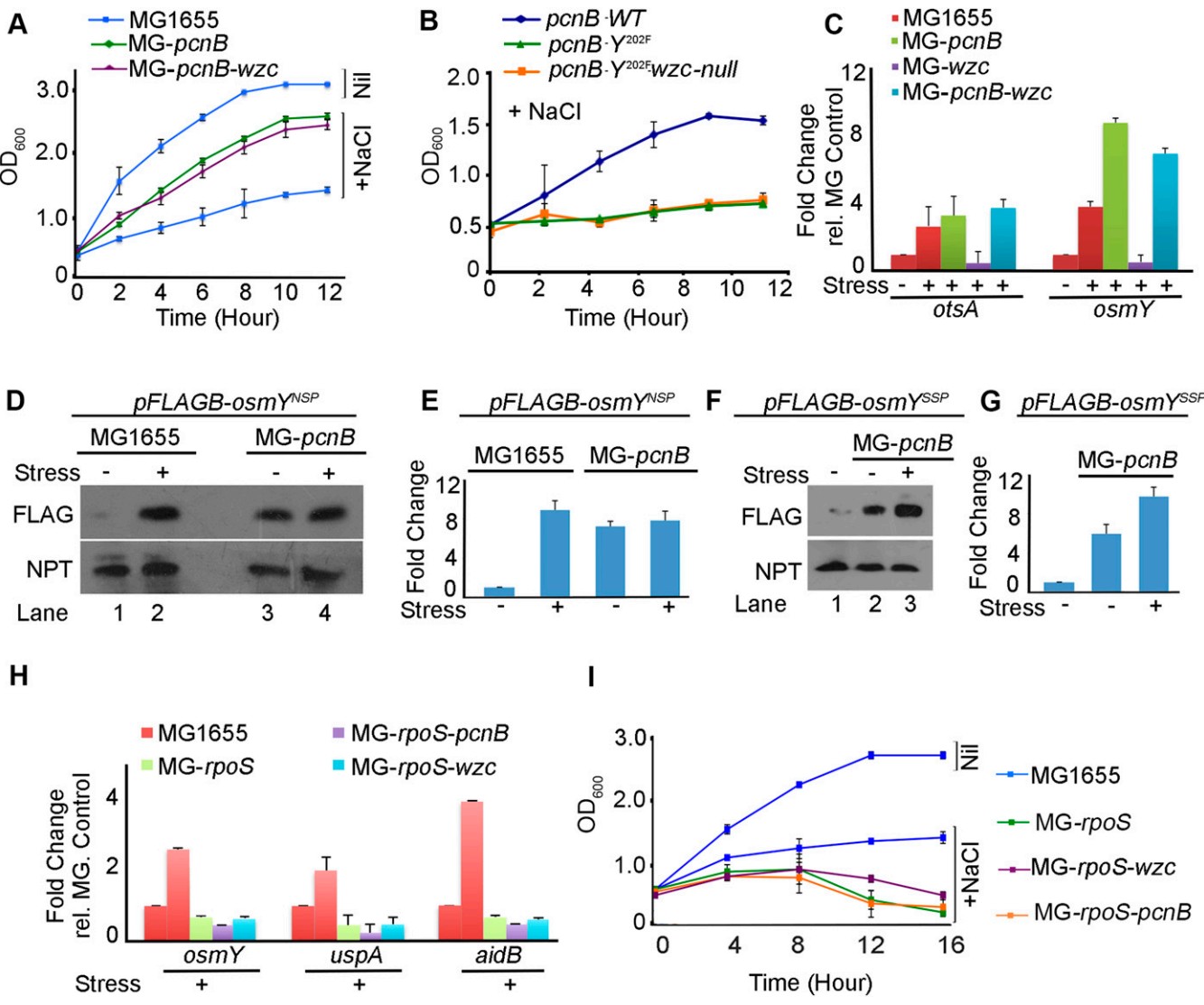

**Figure 5. PAPI activity–regulated stress response gene expression is distinct and operates downstream of RpoS transcriptional induction.**
**(A)** Growth curve analysis of MG1655, MG-*pcnB*, and MG-*pcnB-wzc* mutant strains in the presence of stress treatment (NaCl). Error bar represents the SEM of n = 3 independent cultures. **(B)** Effect of *wzc*-null mutation in the growth curve after stress (NaCl) treatment of MG-*pcnB* strain expressing mutant FLAG[B]-*pcnB-Y202F*. Control growth curve of WT FLAG[B]-*pcnB (WT)*–expressing cells is indicated. **(C)** qRT–PCR analysis of PAPI target mRNAs from total RNA isolated from strains as in MG1655, MG-*pcnB*, and MG-*pcnB-wzc* and MG-*wzc* cells in the presence and absence of stress (NaCl) exposure as indicated. (*P* < 0.001 for both *osmY* and *otsA*). **(D)** Western blot analysis of FLAG-OsmY reporter protein expression (using anti-FLAG antibody) after transformation of pFLAG[B]-*osmY*[NSP] (OsmY expressed from stress-insensitive promoter and *osmY* UTR) plasmid in MG1655 and *pcnB*-null (MG-*pcnB*) strains. Plasmid-expressed neomycin phosphotransferase is shown as a loading control. **(E)** qRT–PCR analysis of reporter expression using a pair of primers (forward from encoding *FLAG* region and reverse from *osmY cds*) for *FLAG-osmY* under conditions as in (D) (*P* < 0.001 for all conditions). **(F)** Western blot analysis of FLAG-OsmY reporter protein expression (using anti-FLAG antibody) after transformation of pFLAG[B]-*osmY*[SSP] (OsmY expressed from stress-sensitive promoter and *osmY* UTR) in MG1655 and *pcnB*-null (MG-*pcnB*) cells. Neomycin phosphotransferase encoded from plasmid is shown as control. **(G)** qRT–PCR analysis of the same reporter expression as in (F). **(H)** qRT–PCR analysis of stress response mRNAs (*osmY*, *aidB*, and *uspA*) from MG1655, MG-*rpoS*, MG-*rpoS-wzc*, and MG-*rpoS-pcnB* mutants in the presence and absence of stress (NaCl) treatment. Error bar represents the SEM of n = 3 independent experiments (*P* < 0.03 for *osmY*, <0.04 for *uspA*, and <0.002 for *aidB*). **(I)** Growth curve analysis of strains as in (H) after stress (NaCl) exposure. Error bar represents the SEM of n = 3 independent cultures.

concert with the master regulator RpoS-mediated transcriptional induction (Battesti et al, 2011; Gottesman, 2019). Controlling polyadenylation that initiates 3′-exonucleolytic mRNA degradation is an efficient way to regulate mRNA turnover (Laalami et al, 2014). Reducing mRNA turnover will prevent the futile cycle of transcriptional stimulation that is countered by the degradation pathway (Nouaille et al, 2017). In addition, mRNA turnover rate determines the steady state level of an mRNA in the cell affecting its cellular availability for translation (Petersen, 1992; Iost & Dreyfus, 1995). Therefore, not only the synthesis of the transcripts but also mRNA turnover is critical to control cellular mRNA available for translation (Petersen, 1992; Iost & Dreyfus, 1995). Thus, the PAPI-mediated stress response pathway will function downstream but cooperatively of the RpoS-mediated transcriptional pathway ultimately to generate an increase in and rapid production of stress response proteins.

It has been shown that cellular polyadenylation level is growth rate–dependent and that PAPI level is maintained low through transcriptional and translational mechanisms (Cao & Sarkar, 1997; Binns & Masters, 2002; Jasiecki & Wçgrzyn, 2003; Jasiecki & Węgrzyn, 2006a, 2006b; Nadratowska-Wesolowska et al, 2010). However, during the cellular growth under stress, there is no change in the *pcnB* mRNA expression in spite of the presence of two σS-dependent promoters (Nadratowska-Wesolowska et al, 2010). Rather, PAPI polyadenylation activity is regulated through phosphorylation that gives growth advantage to the cell. We have shown that PAPI activity is reduced through tyrosine phosphorylation at the catalytic domain phenotypically making it similar to *pcnB*-null mutation. Therefore, in this stress response mechanism, the stress signal primarily acts through phosphorylation. This phosphorylation could putatively function as a signalling cascade where a tyrosine kinase phosphorylates and deactivates its downstream negative mediator protein PAPI that in turn allows stimulated expression of stress response genes (Fig S6). This phosphorylation could regulate PAPI membrane localisation and its assembly into ribonucleoprotein bodies in the cell (Jasiecki & Węgrzyn, 2005). This will help in signal transmission and induced translation analogous to that of eukaryotic stress granules to counteract minor changes in the environment (Buchan & Parker, 2009; Muthunayake et al, 2020).

Bacterial tyrosine kinases are well known in regulating extracellular polysaccharide synthesis, cell division, stress response, and pathogenesis (Grangeasse et al, 2003, 2012; Obadia et al, 2007; Jittawuttipoka et al, 2013). In this study, we identify PAPI as a novel target of Wzc tyrosine-protein kinase. Wzc tyrosine kinase regulates the biosynthesis of capsular polysaccharide and exopolysaccharide production and export in *E. coli* (Obadia et al, 2007). Our study establishes a role of Wzc in stress response gene expression through PAPI phosphorylation downstream of a stress signal (Fig S6). Although PAPI activity is controlled through Wzc kinase activity, Wzc activity is apparently not affected during stress response, and instead, its mRNA expression is induced. The exact mechanism of how *wzc* expression is induced during stress is unclear. *wzc* has two promoters: one dependent on the sigma factor RpoE (σ24), and the other dependent on the general sigma factor RpoD (Stout, 1996; Huerta & Collado-Vides, 2003; Rhodius et al, 2006). RpoE is known to regulate transcription of genes involved in response to growth in stress conditions such as high temperature, hyperosmotic shock, or unfolded protein response (Rowley et al, 2006). Moreover, the RpoE-dependent promoter on the *wzc* gene is regulated by the transcriptional activator RcsAB of the Rcs-phosphorelay system, a well-established regulator of envelope stress response (Wehland & Bernhard, 2000; Hagiwara et al, 2003). Thus, RpoE could function with RcsAB to regulate *wzc* transcription that mediates general stress response in *E. coli* through the Wzc-PAPI phosphorylation system.

Moreover, the exact mechanism of how PAPI tyrosine phosphorylation reduces PAPI activity is unclear. Phosphorylation often changes protein structural properties affecting its activity, stability, or interacting partners (Cohen, 2000). Tyr-202 in the PAPI catalytic domain is in the nucleotide interaction region in the M3 motif (Raynal & Carpousis, 1999; Betat et al, 2004). Our simulation studies indicate that Y202 phosphorylation could induce structural changes in PAPI. This structural alteration may disrupt the catalytic

active site affecting ATP binding leading to reduced polyadenylation activity. Similar effects of phosphorylation have also been reported in *E. coli* isocitrate dehydrogenase enzyme where a threonine phosphorylation decreases activity of the enzyme through reduced substrate binding (Thorsness & Koshland, 1987; Cozzone & El-Mansi, 2005). Likewise, heat shock response sigma factor (RpoH) is phosphorylated that renders the protein inactive (Klein et al, 2003; Iyer et al, 2020). Phosphorylation of RNA polymerase β′ subunit by the phage-encoded kinase Gp0.7 inhibits its elongation activity and induces transcription termination during bacteriophage T7 infection (Severinova & Severinov, 2006). However, unlike PAPI, many of this phosphorylation is at the serine or threonine residues. Our study for the first time identified a physiologically relevant PAPI tyrosine phosphorylation in vivo that regulates general stress response in *E. coli*. It remains to be explored whether the same phosphorylation affects other cellular aspects of PAPI function including PAPI cellular localisation or assembly into the degradosome complex.

# Materials and Methods

## Bacterial strains and growth conditions

All experiments were carried out in *E. coli* K12 strain MG1655 background. Genotypes of strains employed in the study are shown in Table 1. DH5α strain was used for cloning and plasmid amplification, whereas BL21(DE3) was employed for recombinant protein expression. *pcnB*-, *bipA*-, *wzc*-, *etK*-, and *ydiB*-null mutant strains and pCP20 plasmids were kind gift from Dr. J Gowrishankar, CDFD. Mutants from different bacterial backgrounds were transferred to MG1655 by P1 transduction as described previously (Thomason et al, 2007). To construct double mutants of MG-*rpoS-pcnB*, MG-*rpoS-wzc*, or MG-*pcnB-wzc*, kanamycin cassettes from the MG-*pcnB* or MG-*rpoS* cells were removed using pCP20 plasmid through FLP recombination followed by P1 transduction from MG-*wzc* and MG-*rpoS* strains (Cherepanov & Wackernagel, 1995; Ellermeier et al, 2002). The kanamycin cassette–stripped MG-*pcnB* cells were also used for transforming with kanamycin-resistant plasmids wherever necessary.

All bacterial strains were grown at 37°C in nutrient-rich Luria–Bertani growth media. Unless otherwise indicated, antibiotics were supplemented at the concentration of 100 μg/ml for ampicillin, 50 μg/ml for kanamycin, and 50 μg/ml for spectinomycin as described previously (Anupama et al, 2011).

## DNA constructs

The *pcnB* gene was PCR-amplified from *E. coli* chromosomal DNA and cloned in the NcoI and XhoI sites of pET21d (pET-*pcnB*). A new plasmid expression vector to obtain a FLAG epitope–tagged bacterial expression (pFLAG[B]) was generated by modifying the mammalian expression vector pCMV-Tag2A. The CMV promoter region was modified into a bacterial promoter by site-directed mutagenesis and inverse PCR strategy. We inserted −35 (TTGACA) and −10 (TATATT) sequences of the pTAC promoter while keeping a length of

**Table 1.  List of strains and plasmids used in the study.**

| Strain | Genotype |
|---|---|
| DH5α | F⁻ endA1 glnV44 thi 1 recA1 relA1 gyrA96 deoR nupG purB20 φ80dlacZΔM15 Δ(lacZYA–argF)U169, hsdR17($r_K^-m_K^+$), λ⁻ |
| BL21(DE3) | E. coli str. B F⁻ ompT gal dcm lon hsdS_B($r_B^-m_B^-$) λ(DE3 [lacI lacUV5-T7p07 ind1 sam7 nin5]) (malB⁺)_{K-12}(λ^S) |
| MG1655 | K-12 F⁻ λ⁻ ilvG⁻ rfb-50 rph-1 |
| MG-pcnB | MG1655 ΔpcnB759::kan |
| MG-bipA | MG1655 ΔbipA733::kan |
| MG-etk | MG1655 Δetk-725::kan |
| MG-wzc | MG1655 Δwzc-758::kan |
| MG-ydiB | MG1655  ΔydiB766::kan |
| MG-rpoS | MG1655ΔrpoS746::kan |
| MG-rpoS-pcnB | MG-rpoS ΔpcnB759::kan |
| MG-rpoS-wzc | MG-rpoS Δwzc-758::kan |
| MG-pcnB-wzc | MG-pcnB Δwzc-758::kan |
| List of plasmids | |
| pBAD18 | Expression vector, pKK223-3 derivative, arabinose-inducible expression, pBR322 origin, ampicillin resistance |
| pET21d | Expression vector, pBR322 origin, N- and C-terminal His-tag on the expressed protein, ampicillin resistance |
| pCL1920 | Cloning vector, pSC101 origin, spectinomycin resistance |
| pFLAG^B | Expression vector, pCMV-Tag2A derivative with bacterial promoter, ribosome-binding site and transcription terminator. N-terminal FLAG epitope tag on the expressed protein, kanamycin resistance |
| pFLAG^B-pcnB | Derivative of pFLAG^B with cloned E. coli pcnB gene in the BamHI and EcoRI sites |
| pET-pcnB | Derivative of pET21d, with cloned E. coli pcnB gene in the NcoI and XhoI sites |
| pFLAG^B-pcnB-Y60F | Derivative of pFLAG^B-pcnB with mutation of Y60F by site-directed mutagenesis |
| pFLAG^B-pcnB-Y169F, Y170F | Derivative of pFLAG^B-pcnB with mutation of Y169F, 170F by site-directed mutagenesis |
| pFLAG^B-pcnB-Y202F | Derivative of pFLAG^B-pcnB with mutation of Y202F by site-directed mutagenesis |
| pFLAG^B-pcnB-Y341F | Derivative of pFLAG^B-pcnB with mutation of Y341F by site-directed mutagenesis |
| pET-pcnB-Y60F | Derivative of pET-pcnB with mutation of Y60F by site-directed mutagenesis |
| pET-pcnB-Y169F, Y170F | Derivative of pET-pcnB with mutation of Y169, 170F by site-directed mutagenesis |
| pET-pcnB-Y202F | Derivative of pET-pcnB with mutation of Y202F by site-directed mutagenesis |
| pET-pcnB-Y341F | Derivative of pET-pcnB with mutation of Y341F by site-directed mutagenesis |
| pFLAG^B-osmY^NS | Derivative of pFLAG^B with cloned E. coli osmY gene in the BamHI and EcoRI sites |
| pFLAG^B-osmY^SS | Derivative of pFLAG^B having modification in the −35 and transcription start site sequence with cloned E. coli osmY gene in the BamHI and EcoRI sites |
| pFLAG^B-wzc | Derivative of pFLAG^B with cloned E. coli wzc gene in the BamHI and EcoRI sites |
| pFLAG^B-WzcK^540R | Derivative of pFLAG^B-wzc with mutation of K540R by site-directed mutagenesis |

17 nucleotides between −10 and −35 sequences (de Boer et al, 1983). A ribosome-binding site (AGAAGG) was then inserted two nucleotides downstream of the transcription start site and six nucleotides upstream of FLAG epitope tag translation initiation codon (ATG) while retaining the multiple cloning site of the Tag2A vector. In addition, E. coli transcription termination sequence (GCCGCCAGTTGCGCT GGCGGCATTTTT) (Platt, 1981) was introduced downstream of the multiple cloning site through inverse PCR strategy. PCR-amplified pcnB was cloned at the BamHI and EcoRI sites of the pFLAG^B vector to generate pFLAG^B-pcnB. pcnB mutations (Y60F, Y169F, Y170F, Y202F, and Y341F) were introduced on pFLAG^B-pcnB and pET-pcnB plasmids by site-directed mutagenesis. The osmY gene with its UTR region of ~116-bp downstream of stop codon was

PCR-amplified from E. coli chromosomal DNA and cloned in the EcoRI and BamHI sites of pFLAG^B to generate a stress-insensitive pFLAG^B-osmY (pFLAG^B-osmY^NS-P) reporter construct. Furthermore, to generate a control stress–sensitive pFLAG^B-osmY (pFLAG^B-osmY^SS-P) reporter construct, two mutations in the −35 region (AATTGACA to TCCCGAGCGA) and after −10 sequence (insertion of TAACAAA) were introduced in the pFLAG^B-osmY promoter to make it a cognate osmY promoter (Yim et al, 1994; Becker & Hengge-Aronis, 2001). The gene for bacterial tyrosine kinase wzc was PCR-amplified from E. coli chromosomal DNA and cloned in the EcoRI and BamHI sites of pFLAG^B (pFLAG^B-wzc) to obtain a FLAG epitope–tagged protein expression. wzc point mutation (K540R) was introduced on pFLAG^B-wzc by site-directed mutagenesis. The list of primers used

for PCR amplification, site-directed mutagenesis, inverse PCR, and cloning is shown in Table S3 in the Supplemental Data 1.

## Induction of stress response and cellular analysis

For stress sensitivity assay, cells were grown to approximately O.D$_{600}$ of 0.5 in LB media. To induce different stress responses, cell culture was treated with 400 mM NaCl (osmotic shock), 5 mM hydrogen peroxide, H$_2$O$_2$ (oxidative stress), 25 μg/ml methyl nitrosoguanidine, MNNG (DNA alkylation stress or DNA damage), low pH at 3.0 adjusted with 100% hydrochloric acid, HCl (acid shock), heat shock at 42°C (heat stress), and cold shock at 4°C (cold stress) for 1 h each as described earlier (Rojas et al, 2014).

For dilution plating experiments, after 1 h of stress treatment cells were serially diluted from $10^{-1}$ to $10^{-6}$ in LB and spotted around 5 μl of diluted cultures on an LB agar plate and incubated at 37°C. For viable colony counting, ~$10^{-4}$ dilution cells were spread on a corresponding selection LB agar plate. Then, viable colonies were counted and expressed as CFU/ml of cell culture. For, molecular experiments, RNA or proteins were tested 1 h post-stress treatment.

For growth curve analysis, cells from the mid-log phase (~0.5 O.D$_{600}$) were treated with respective amounts of stressors as described above and OD$_{600}$ readings were followed at every hour until the cultures reached the stationary phase. The O.D$_{600}$ at the time of addition of stressors was considered as initial (0) time point, and a graph was plotted with OD$_{600}$ versus time in hours.

## Quantitative real-time PCR (qRT–PCR) and half-life measurement

Bacterial cells were grown at 37°C and harvested at OD$_{600}$ ~0.6, and total RNA was isolated by the TRIzol method as described earlier (Sudheesh & Laishram, 2017) and quantified by A$_{260}$ measurement. 2.5 μg of total RNA was used to synthesise first strand cDNA with random hexamers and MMLV reverse transcriptase. qRT–PCR was performed with gene-specific primers and quantified with the CFX96 multi-colour system using SYBR Green Supermix (Bio-Rad) as described previously (Sudheesh et al, 2019). Single-product amplification was confirmed by melt-curve analysis, and primer efficiency was near 100% in all experiments. Quantification is expressed in arbitrary units, and target mRNA abundance was normalised to the expression of *dxs* or *rrsA* with the Pfaffl method as described earlier (Pfaffl, 2001).

For the half-life measurement of mRNA, cells were treated with 500 μg/ml rifampicin to inhibit transcription and harvested at multiple time points (0–24 min) post-rifampicin treatment as described earlier (Selinger et al, 2003). RNA was isolated from each cell collected from each of the time points (0, 1, 2, 4, 6, 8, 12, 16, and 24 min). qRT–PCR was carried out, and half-life (T½) was measured as described earlier by following the decrease in % mRNA level over time with 0 time point taken as 100% of each gene expression (Sudheesh et al, 2019).

## 3′-RACE assay

Total RNA was isolated from MG1655 and MG-*pcnB* strains in the presence and absence of one representative stress (NaCl) treatment using TRIzol reagent as described in the manufacturer's

instructions. First strand cDNA was synthesised using MMLV reverse transcriptase and 3 μg of total RNA with an oligo-dT primer having an adapter sequence at the 5′-end (adapter primer) as described previously (Sudheesh et al, 2019). cDNA was further amplified using gene-specific forward primer and the primer specific to the 5′-end adapter sequence (AUAP reverse primer). Sequences of adapter primer, AUAP primer, and gene-specific primers are shown in the Supplemental Data 1.

## Plasmid copy-number analysis

To analyse the plasmid content, MG1655 or MG-*pcnB* strains were transformed with colE1-based pBAD18 plasmid and another *RNAI*-independent non-colE1 plasmid pCL1920. From the cultures collected after the treatment with different stresses or control stress–untreated cultures in LB medium, plasmids were isolated by alkaline lysis, digested with NdeI enzyme, and analysed on a 1% agarose gel as described earlier (Whelan et al, 2003).

## In vitro polyadenylation assay

In vitro polyadenylation assay was carried out using recombinant His-PAPI or purified FLAG-PAPI on a universal (UAGGGA)$_5$A$_{15}$ (A$_{45}$) template with radiolabelled [α-$^{32}$P]-ATP in a PAP assay buffer (250 mM NaCl, 50 mM Tris–HCl, and 10 mM MgCl$_2$, pH 7.9 at 37°C) as described earlier (Mellman et al, 2008). RNA products were analysed on a 6% urea-denaturing polyacrylamide gel and visualised by phosphor imaging. For stress inductions, pFLAG$^B$-*pcnB*–expressed cells were treated with appropriate stress inducers, and FLAG-PAPI was immunoprecipitated using anti-FLAG agarose beads (Sigma-Aldrich). PAPI proteins were then eluted from the beads using 3×-FLAG peptide in IP dilution buffer containing 20 Mm Tris–HCl, 150 mM NaCl, 1% Triton X-100, 2 Mm EDTA, and 0.01% SDS. PAP assays were carried out using 2 μg of the eluted proteins. In the case of PAP assay with cell lysates, ~10 μg total equivalent cell lysates was used for each PAP assay reaction. For oligo-(dT)/RNase H digestion experiment, digestion of the poly(A) RNA was carried out after the in vitro polyadenylation reaction in 200 mM KCl, 1 mM EDTA, 20 mM Tris–HCl, pH 8.0, 30 mM MgCl$_2$, and 20 U RNase inhibitor as described earlier (Mellman et al, 2008). ~5 nM oligo-(dT) was annealed to the RNA primer, and digestion was performed at 37°C for 60 min with 4 U of RNase H (Ambion). For phosphatase treatment of FLAG-PAPI protein, immunopurified FLAG-PAPI protein was incubated with 20 units of λ-protein phosphatase (New England Biolabs) or YopH enzyme (MyBioSource) at 30°C for 30 min in the presence or absence of 12 mM sodium vanadate and its corresponding buffer in the presence of 1 mM MnCl$_2$.

## Protein purification

Recombinant proteins were expressed using pET21-d plasmid constructs, overexpressed in BL21(DE3) by inducing with 1 mM isopropyl thio-β-D-galactoside at 18°C. Cells were lysed in ice-cold lysis buffer (20 mM Tris and 200 mM NaCl) and purified by Ni-NTA affinity chromatography as described previously (Sudheesh et al, 2019). The purified proteins were dialysed in protein storage buffer

(20 mM Tris–HCl and 100 mM NaCl), concentrated using polyethylene glycol (PEG 20000 mw), snap-frozen, and stored at −80°C.

## Immunoprecipitation (IP) and immunoblotting

IP was carried out from the cell lysates prepared from 10 ml bacterial culture of $OD_{600}$ 1.00 in lysis buffer (20 mM Tris [pH 8.0], 150 mM KCl, 1 mM $MgCl_2$, and 1 mM DTT). ~4 mg of total protein was used for each IP experiment as described previously (Mohan et al, 2018). For Immunoblotting experiments, cell lysates or IP eluates were resolved on a SDS–PAGE and blotted using specific antibodies on a PVDF membrane as described earlier (Mohan et al, 2018). In IP experiments, input was loaded at an amount that is equivalent of 10% of the IP sample. Immunopurification of FLAG-PAPI was carried out with anti-FLAG affinity agarose and eluted using 3X-FLAG peptides (Sigma-Aldrich) as described previously (Mohan et al, 2015).

## In vitro kinase assay

Recombinant His-PAPI or FLAG-PAPI (10 $\mu$g) was incubated with 10 $\mu$Ci [$\gamma^{32}$P] ATP in the presence of 10 $\mu$g protein equivalent of cell lysate in the kinase buffer (250 mM Tris–HCl, pH 7.5, 50 mM glycerol phosphate, 100 mM $MgCl_2$, 0.1 mM sodium vanadate, and 10 mM DTT) as described earlier (Grangeasse et al, 2003; Mellman et al, 2008). For in vitro kinase assay using FLAG-Wzc or FLAG-WzcK540R, Wzc was immunoprecipitated using anti-FLAG agarose beads (Sigma-Aldrich) and eluted from the beads using 3X-FLAG peptide in IP dilution buffer (20 mM Tris–HCl, 150 mM NaCl, 1% Triton X-100, 2 mM EDTA, and 0.01% SDS). For in vitro kinase assays using cell lysates, bacterial cell pellets after respective stress treatment were resuspended in lysis buffer containing 100 mM Tris–Cl (pH 7.5) and 50 mM $\beta$-mercaptoethanol. Cells were then lysed by sonication at 4°C for 5-s ON and 5-s OFF cycles for 2 min. Reactions were incubated at 37°C for 1 h and terminated by the addition of SDS sample dilution buffer. The kinase reaction was analysed on SDS–PAGE and detected by phosphor imaging. For in vitro kinase reactions using FLAG-purified PAPI protein, FLAG-PAPI was heat-inactivated by incubation at 65°C for 15 min before kinase assay reaction as described earlier (Mellman et al, 2008). Whenever in vitro kinase reactions were followed by downstream phosphatase treatment or polyadenylation assays, buffers were supplemented with required salts to maintain the appropriate buffer (1X) composition.

## Phosphoamino acid analysis by one-directional TLC

PAPI after in vitro kinase reaction was hydrolysed by mixing with 10× vol of 6 N HCl in a tightly capped tube at 110°C for 1 h. Samples were then spun down, and the remaining HCl was evaporated at 110°C. Samples were then resuspended in 5 mM KOH before loading onto TLC. Alongside, an equivalent amount of hydrolysed standard phosphoamino acid (phosphotyrosine or phosphoserine, 1 $\mu$g) was also spotted in a 20 × 20 cm TLC plate as described earlier (Tavare et al, 1991; Islas-Flores et al, 1998). Samples were then separated on a TLC buffer system having 5:2:1 butanol: formic acid: water until it migrated to the top edge of the TLC plate. TLC plates were then sprayed with 0.1% solution of ninhydrin in acetone, air-dried and later scanned for standard amino acids, and exposed to analysis on a phosphor screen for radiolabelled hydrolysed amino acids.

## Molecular dynamics (MD) simulations

To unravel the Y202 phosphorylation–induced structural changes in PAPI, we employed PAPI crystal structure in the presence and absence of ATP interaction. The atomic coordinates of the WT PAP enzyme in its apo form were retrieved from the protein data bank (PDB ID: 3AQK) (Toh et al, 2011). The missing residue coordinates (114–137) were modelled using the MODELLER 10.2 software package (Webb & Sali, 2014). Then, we carried out a 300-ns MD simulation with the apo state of the enzyme (PDB ID: 3AQK). Residue Y202 was phosphorylated to mimic the phosphorylated PAP enzyme using the Vienna-PTM server (Margreitter et al, 2013). H++ server and PROPKA 3.0 were used to identify the protonation states of the residues (Bas et al, 2008; Anandakrishnan et al, 2012). The systems were solvated with TIP3P water molecules in a 10 Å cubic box (Jorgensen, 1983). To neutralise the systems and maintain the intracellular ionic strength at 150 mM, K+ and Cl− ions were used.

All simulations were carried out using the GROMACS 2022.3 biomolecular simulation software package with periodic boundary conditions (Pronk et al, 2013). The CHARMM36m force field was used, and the particle-mesh Ewald sum method was used to model the long-range electrostatic interactions with the direct sum cut-off of 10 Å and the Fourier spacing of 1.2 Å. Bond lengths were constrained via the LINCS algorithm, and van der Waals interactions were treated with a 10 Å cut-off (Evans & Holian, 1985; Essmann et al, 1995; Huang et al, 2017). Initially, energy minimisation of the systems was done using the steepest descent energy minimisation method. Using Nose–Hoover thermostat, systems were equilibrated for 5 ns each in the NVT (constant particle number, volume, and temperature) and NPT (constant particle number, normal pressure, and temperature) ensembles (Evans & Holian, 1985). As specified, systems were minimised and equilibrated to a temperature of 303.15 K. Then, systems were subjected to 300 ns of final production of MD runs at 303.15 K in an NPT ensemble. During simulations, the coordinates were stored every 100 picoseconds for analyses.

## Genome-wide RNA-seq analysis

RNA extracted from pellets of MG1655 and MG-*pcnB* was analysed for QC using Bioanalyser and Qubit. Library preparation and deep sequencing was performed at commercially available genomics facility at the Genotypic Technology (https://www.genotypic.co.in). 4 $\mu$g of QC passed total RNA was used for ribodepletion using RiboMinus Bacterial Kit (Invitrogen). Furthermore, 100 ng of Qubit quantified ribodepleted RNA was taken for transcriptome library preparation according to the SureSelect Strand-Specific RNA Library Prep Kit protocol outlined in "SureSelect Strand-Specific RNA Library Prep for Illumina Multiplexed Sequencing" (Illumina). Briefly, the RNA was fragmented for 4 min at 94°C in the presence of divalent cations and first strand cDNA was synthesised. The single-stranded cDNA was cleaned up using High Prep (Cat # AC-60050; Magbio). Strand specificity was maintained by the addition of actinomycin D. Second strand cDNA was synthesised and end-repaired using Second Strand Synthesis using End-Repair Mix.

The cDNA was cleaned up using High Prep (Cat # AC-60050; Magbio). Adapters were ligated to the cDNA molecules after the addition of "A" base. High Prep cleanup was performed post-ligation. The library was indexed and enriched for adapter-ligated fragments using 10 cycles of PCR. The prepared library was quantified using Qubit and validated for quality by running an aliquot on High Sensitivity Bioanalyzer Chip (Agilent). 150 (75 × 2)-bp paired-end sequencing was carried out on Illumina platform to generate 20–25 million PE reads per sample.

Bioinformatics analysis was performed with Alignment Statistics, Reference alignment with depth statistics, digital gene expression, and DESeq plot counts. The raw data generated were checked for the quality with FastQC1, and pre-adapter sequences and low-quality bases were removed. It was aligned to *E. coli* (K-12) (https://bacteria.ensembl.org/Escherichia_coli_str_k_12_substr_mg1655_gca_000005845/Info/Index/) using TopHat-2.0.133 (Trapnell et al, 2009) reference genome. Transcript assembly was done using Cufflinks-2.2.1 (Trapnell et al, 2010), which assembles transcripts, estimates their abundance, and tests for differential expression and regulation in RNA-Seq samples. Then, using cuff merge, Cufflinks assemblies were combined followed by differential gene expression analysis using "Cuffdiff" to obtain the significant changes in transcript expression (Trapnell et al, 2010). UniProt knowledge base was used to annotate the genes for gene ontology (Boutet et al, 2012). Differential gene expression studies were carried out between MG-*pcnB* group versus WT MG1655 samples. Differentially expressed genes were further annotated for the protein name and gene ontology, and heat map was generated as a representation of the expression values (as colours). The raw RNA-Seq data of our study have been deposited in the NCBI GEO sequencing data repository.

### In silico prediction of phosphorylation sites and promoter targets of RpoS/RpoD

Putative tyrosine phosphorylation sites were predicted using NetPhos 3.1 software (http://www.cbs.dtu.dk/services/NeTPhos/) (Blom et al, 1999). Target PAPI primary sequence was uploaded as input and analysed for available tyrosine phosphorylation sites with a minimum threshold score at 0.5. The promoter sequence of the *wzabc* operon was identified using RegulonDB, and the corresponding sequence is marked and labelled with respect to the corresponding transcription start site (Tierrafria et al, 2022).

### Statistics

All data were obtained from at least three independent experiments and are represented as the mean ± SEM. The statistical significance of the differences in the mean is calculated using ANOVA with statistical significance at a *P*-value less than 0.05. All Western blots show representative of at least three independent blotting experiments.

### Primers and antibodies

The list of all the primers and antibodies employed in the study is shown in the Supplemental Data 1.

## Data Availability

The data discussed in this publication have been deposited in NCBI's Gene Expression Omnibus and are accessible through GEO Series accession number GSE215029 (https://www.ncbi.nlm.nih.gov/geo/query/acc.cgi?acc=GSE215029).

## Supplementary Information

## Acknowledgements

Null mutant strains (*pcnB*, *etk*, *wzc*, *bipA*, and *ydiB*) were kind gift from Dr. J Gowrishankar, Centre for DNA Fingerprinting and Diagnostics, Hyderabad. This work was supported by grants from the Department of Biotechnology (BT/PR15554/BRB/101464/2015) and Swarnajayanti Fellowship from the Department of Science and Technology (DST/SJF/LSA-03/2018-19), Government of India to RS Laishram, Indian Council of Medical Research (ICMR) Research Fellowships to N Francis, and UGC JRF to MR Behera.

### Author Contributions

N Francis: conceptualisation, data curation, formal analysis, validation, investigation, visualisation, methodology, and writing—original draft, review, and editing.
MR Behera: validation, investigation, visualisation, methodology, and writing—review and editing.
K Natarajan: data curation, software, validation, and visualisation.
RS Laishram: conceptualisation, resources, data curation, formal analysis, supervision, funding acquisition, validation, investigation, visualisation, methodology, project administration, and writing—original draft, review, and editing.

### Conflict of Interest Statement

The authors declare that they have no conflict of interest.

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
