## [Reviewer comments · Life Science Alliance]

Life Science Alliance

Tyrosine phosphorylation controlled poly(A)polymerase activity regulates stress response in bacteria

Nimmy Francis, Malaya R. Behera, Kathiresan Natarajan, and Rakesh S. Laishram

DOI: <https://doi.org/10.26508/lsa.202101148>

Corresponding author(s): *Dr. Rakesh S. Laishram (Rajiv Gandhi Centre for Biotechnology)*

Review Timeline:

Submission Date:	2021-06-30
Editorial Decision:	2021-08-20
Revision Received:	2022-10-09
Editorial Decision:	2022-11-04
Revision Received:	2022-11-28
Accepted:	2022-11-29

Scientific Editor: Novella Guidi

Transaction Report:

August 20, 2021

Re: Life Science Alliance manuscript #LSA-2021-01148-T

Dr. Rakesh S. Laishram
Rajiv Gandhi Centre for Biotechnology
Cancer Research program
Thycaud Post, Poojappura
Trivandrum 695014
India

Dear Dr. Laishram,

Thank you for submitting your manuscript entitled "Tyrosine phosphorylation controlled poly(A)polymerase activity regulates stress response in bacteria" to Life Science Alliance. The manuscript was assessed by expert reviewers, whose comments are appended to this letter. Both reviewers are quite positive about the work and think that it is novel and of interest, and this mechanism seems likely to play an important part in the response of *E. coli* to stress. In their view the genetic evidence for these conclusions is quite strong, but the biochemical evidence that stress treatment results in the inactivation of PAP1/pcnB through phosphorylation of Tyr202 by the Wzc BY kinase is incomplete, and needs to be strengthened with additional experiments, quantifications and inclusion of controls. The relatively weak phosphorylation signals either as pTyr blots or as radioactive bands, cast doubts that the phosphorylation is indeed on PAPI. Also, they both point out that the Wzc phosphorylation data are not totally convincing. In fig 4 panel F, the pTyr-positive bands in the upper panel are rather faint, and need quantifying. Reviewer 2 suggests to somewhat reduce the concentration of unlabeled ATP (currently 50 μ M) in their reactions to increase signal intensity. All the other concerns raised by the reviewers should be addressed as well. We, thus, encourage you to submit a revised version of the manuscript back to LSA that responds to all of the reviewers' points.

Thank you for this interesting contribution to Life Science Alliance. We are looking forward to receiving your revised manuscript.

Sincerely,

-- Summary blurb (enter in submission system): A short text summarizing in a single sentence the study (max. 200 characters

including spaces). This text is used in conjunction with the titles of papers, hence should be informative and complementary to the title and running title. It should describe the context and significance of the findings for a general readership; it should be written in the present tense and refer to the work in the third person. Author names should not be mentioned.

B. MANUSCRIPT ORGANIZATION AND FORMATTING:

Reviewer #1 (Comments to the Authors (Required)):

The *E. coli* poly(A) polymerase I (PAPI), which is encoded by the *pcnB* gene, adds poly A to the 3' ends of target RNAs triggering their destabilization, and here the authors have investigated how the activity of PAPI is regulated in response to stress. They started by using RNA-seq to identify RNAs that were stabilized in a *pcnB*-null MG1655 strain compared to WT MG1655, finding ~600 RNAs whose levels were increased, of which ~170 RNAs represented a subset of RNAs known to be involved in stress responses. On this basis, the authors tested whether PAPI is required for *E. coli* stress responses, showing that *pcnB*-null cells were more resistant to six different stress treatments - heat, cold, acid, salt, H₂O₂, and DNA damaging agents; in contrast, overexpression of Flag-tagged PAPI rendered WT cells more sensitive to stress treatment, both findings being consistent with PAPI being important for establishing the level of cell stress resistance. By qPCR and RNA half-life analysis they showed that the absence of PAPI resulted in stabilization of several selected stress response mRNAs, such as *osmY*. Using 3' RACE, they also found a decrease in the level of polyadenylated *osmY* RNA in stress treated cells, equivalent to that observed in *pcnB*-null cells. The level of PAPI protein was unaffected by stress treatment, but the total RNA polyadenylation activity in lysates of stress-treated cells or His-tagged PAPI isolated from stress-treated cells was reduced, suggesting that PAPI activity might be negatively regulated as a result of posttranslational modification(s) under stress conditions. To test directly whether PAPI phosphorylation might be induced by stress, they used anti-pSer and anti-pTyr Abs to blot Flag-PAPI isolated from cells treated with all six different stresses, and found an increase in the pTyr signal in PAPI, without a pSer signal being detected under any condition. A pTyr signal was also detected in purified His-tagged PAPI exposed to a lysate from 400 mM NaCl-treated cells incubated in the presence of ATP. Next, they used NetPhos to predict possible sites of Tyr phosphorylation in PAPI, focusing on Y169, Y170, Y202 in the catalytic domain, and Y60 and Y341 in flanking regions. These Tyr were mutated to Phe in Flag-PAPI and His-PAPI, which were expressed in *pcnB*-null cells, and assayed for the presence of pTyr. They found that only the Y202F mutation reduced the level of pTyr in Flag-PAPI isolated from NaCl-treated cells. They also showed phosphorylation of purified His-PAPI proteins incubated in lysates from stress-induced cells reduced its polyadenylation activity, and that λ phosphatase treatment of the immunopurified Flag-PAPI from stress-treated cells increased its polyadenylation activity. Using γ 32P-ATP they found that all the His-PAPI proteins except Y202F His-PAPI were phosphorylated in lysates if stress treated cells. They went on to show that the activity of immunopurified Y202F-Flag-PAPI was higher than that of WT or Y341 Flag-PAPI. From these experiments, they deduced that phosphorylation of Y202 in the catalytic domain of PAPI inhibits its polyadenylation activity. Consistently, stress treatment of Y202Y PAPI expressing cells did not induce an increase in *osmY*, *uspC*, and *otsA* RNAs, and Y202F PAPI cells were hyper-sensitive to stress treatment. Next, they searched for the Tyr kinase that phosphorylates Y202 in PAPI by testing loss of function mutants of four known bacterial Tyr (BY) kinases - BipA, Etk, Wzc and YdiB - for effects on stress-induced PAPI Tyr phosphorylation, and found that only the *wzc*-null mutant cells failed to exhibit increased pTyr in isolated Flag-PAPI upon NaCl treatment. To establish that Wzc is the stress-active PAPI Tyr kinase, they carried out in vitro phosphorylation assays with His-PAPI with stress-treated cell lysates, and showed that the *wzc*-null cell lysate had reduced His-PAPI phosphorylating activity compared to WT cell lysate, as measured by pTyr blotting. Finally, they showed that isolated Flag-Wzc was able to phosphorylate purified His-PAPI in the presence of γ 32P-ATP and that this signal was reduced when the Y202F His-PAPI mutant protein was tested. In a separate line of inquiry, by using *rpoS*-null mutation in combination with *pcnB*-null or *wzc*-null mutants, they showed that the regulation of PAPI activity by Wzc-mediated phosphorylation of Y202, is distinct and downstream from the stress-induced RNA polymerase sigma factor/RpoS-mediated transcriptional response pathway.

The finding that the activity of the *E. coli* poly(A) polymerase I (PAPI) is negatively regulated by stress-induced phosphorylation of Tyr202 in the PAPI catalytic domain upon activation of the Wzc BK family tyrosine kinase is novel and of interest, and this mechanism seems likely to play an important part in the response of *E. coli* to stress. The genetic evidence for these

conclusions is quite strong, but the biochemical evidence that stress treatment results in the inactivation of PAP1/pcnB through phosphorylation of Tyr202 by the Wzc BY kinase is incomplete, and needs to be strengthened for the following reasons.

1. The conclusion that stress-induced inactivation is due to Tyr phosphorylation and not Ser/Thr phosphorylation of PAPI was based on blotting with anti-pSer and anti-pTyr Abs. However, anti-pSer Abs are known to be notoriously poor, and recognize only a subset of pSer phosphorylation sites. In other words, the data in Figure 3A/B do not rule out that Ser/Thr phosphorylation plays a role in the negative regulation of PAPI activity. Since the authors used ³²P-labeling to demonstrate increased phosphorylation of His-PAPI by lysates from stressed cells, they are in a position to carry out a direct phosphoamino acid analysis on ³²P-labeled PAPI to determine whether phospho-His-PAPI from stress-treated cells contains pSer/pThr in addition to pTyr.

2. The authors provide no direct evidence that Tyr202 in PAPI is phosphorylated in response to stress - the effect of the Y202F mutation could be indirect. The generation and use of a pY202 PAPI-specific antiserum would be one way to establish this. Alternatively, they should carry out direct MS-based phosphoproteomic analysis on either Flag-PAPI from stress-treated cells or phospho-His-PAPI incubated in stress-treated cell lysates to identify the stress-induced sites of phosphorylation, i.e., both pTyr and pSer/pThr sites, and demonstrate phosphorylation of Y202. As it stands, while the Y202F mutation results indicate that Y202 is needed for reduction in enzymatic activity, Y202 phosphorylation might not be sufficient.

3. As evidence that phosphorylation of PAPI negatively regulates its enzymatic activity, the authors showed that treatment of PAPI isolated from stress-treated cells with λ phosphatase led to increased polyadenylation activity. They concluded that this increase was due to dephosphorylation of pTyr in PAPI, which resulted in enzymatic reactivation. However, they did not show that λ phosphatase treatment actually caused loss of the pTyr signal, which is essential for them to demonstrate. They also need to include a control in which the λ phosphatase treatment of PAPI is carried out in the presence of a phosphatase inhibitor. Most importantly, λ phosphatase is not specific for pTyr, but can also dephosphorylate pSer/pThr, and so the fact that λ phosphatase treatment increased PAPI polyadenylation activity does not demonstrate that the stress-induced decrease in PAPI activity was due solely to Tyr phosphorylation. To establish this, the authors need to redo this experiment treating PAPI from stress-treated cells with a pTyr-specific phosphatase, such as recombinant YopH, showing that this causes a loss of pTyr in PAPI and also leads to enzymatic reactivation.

4. For stress-induced phosphorylation of Y202 to efficiently inhibit PAPI polyadenylation activity, the stoichiometry of Y202 phosphorylation would have to be high. Did the authors estimate the stoichiometry of PAPI Tyr phosphorylation, either in vivo or in vitro? For instance, if a higher resolution gel is run, is the mobility of PAPI shifted upon phosphorylation and restored when the sample is treated with λ phosphatase?

5. Assuming that the authors' conclusions are correct, there are no insights into how a phosphate linked to Tyr202 would inhibit PAPI catalytic activity - is there a structure of the PAPI catalytic domain or of a related polymerase that could be used to model how Tyr202 phosphorylation might affect poly(A) polymerase activity? D212 and R215 are important catalytic residues - could the phosphate on Y202 interfere with one or both of these residues, for instance, by charge repulsion for D212 or interaction with R215? Is the OH of Y202 predicted to be accessible for phosphorylation by Wzc in the active site?

6. The evidence that Wzc BY tyrosine kinase directly phosphorylates PAPI at Y202 looks reasonable, but Wzc has not been shown convincingly to phosphorylate any protein other than itself, where autophosphorylation of its C-terminus is involved in polysaccharide biosynthesis. Is there a kinase-dead point mutation of Wzc that could be used as another control? If Wzc is the Tyr kinase responsible for phosphorylating PAPI upon stress treatment, how is it activated by stress - for instance, does Wzc exhibit increased autophosphorylation upon stress treatment. Or is Wzc constitutively active, with the level of pY202 in PAPI being controlled by a stress-regulated phosphatase? What is the pTyr phosphatase that dephosphorylates pY202 PAPI and restores its activity, when the stress response is resolved? In this regard, the authors do not show how rapidly the pTyr signal in PAPI is lost when the stressor is removed.

Points: 1. Figure 3A/B: The level of pTyr in PAPI was significant in these samples, but it is not stated in the figure legend whether the cells from which these samples were obtained had been stress treated.

2. Figure 3G: There seems to be no description of how the bacterial cells were lysed and the cell lysates prepared (e.g. what was the buffer and is this compatible with the phosphorylation reactions carried out with the cell lysates?) This may be described elsewhere, but detailed methods need to be provided). Likewise, there is no description of how the lysate phosphorylation assays were carried, i.e. what was the protein concentration, the final concentration of added ATP, incubation time and temperature, etc.

3. Figure 3H: The M1-3 boxes in the schematic need to be defined in the legend?

4. Page 9: The NetPhos prediction algorithm was developed to predict phosphorylation sites in eukaryotic proteins, and there is no reason to believe that this algorithm will be successful in predicting (Tyr) phosphorylation sites in bacterial proteins, where protein phosphorylation is mediated by totally different types of (Tyr) protein kinase to those used in eukaryotes.

5. Figure 3K: The fact that there was no background in the Y3 lane, compared to that observed in WT, Y1, Y2 and Y4 lanes is odd, and needs to be explained. Moreover, based on the shapes of the His-PAPI bands in the lower panel, it does not look as though these are the same as those that were phosphorylated. Exactly how this experiment was carried out needs to be explained in more detail.
6. Figure S3A: As was the case for Figure 3K, it does not look as though the His-PAPI bands in the lower panel are the same as those that were phosphorylated/blotted with anti-pTyr Ab in the upper panel. Why is the upper panel labeled 32P, when the legend says that these samples were blotted with anti-pTyr Abs?
7. Figure S3C: This experiment should include a λ phosphatase control to show that this restores the polyadenylation activity of His-PAPI exposed to the lysate from stress-treated cells, and also include a control in which the λ phosphatase is added to the unphosphorylated His-PAPI protein, and another where a λ phosphatase inhibitor is added.
8. Figure S3D: The right-hand lane where the activity of the dephosphorylated Flag-PAPI was measured has a very strong band that was not obvious in the other assay lanes - do the authors have an explanation? Why did the authors not quantify these three lanes like they did those in panel C? It is essential that the authors repeat this experiment to show that the λ phosphatase treatment completely removed the Flag-PAPI pTyr signal, and also include a control in which the λ phosphatase is added to the unphosphorylated Flag-PAPI protein.
9. Figure 4F and G : These Wzc phosphorylation data are not totally convincing. In panel F, the pTyr-positive bands in the upper panel are rather faint, and need quantifying, and again do not seem to correspond to the His-positive bands in the lower panel. In panel G, the fact that the Y4 and Y3 samples are not in order is confusing, and the very high background in the Y4 lane needs explaining. Also, these bands need quantifying. Did the authors measure the polyadenylation activity of PAPI isolated from wzc-null cells compared to WT cells? Did they show that Flag-Wzc phosphorylation of PAPI reduced its polyadenylation activity?
10. pY202 does not seem to have been identified in published E. coli pTyr phosphoproteomic databases, but this should be checked more thoroughly (the pY202 tryptic peptide may be too small to be identified by MS).

Reviewer #2 (Comments to the Authors (Required)):

Francis et al. propose a novel pathway of regulation of stress response in E.coli, downstream of the master stress regulator RpoS. According to their findings, stress-induced tyrosine phosphorylation of poly (A) polymerase (encoded by pcnB) through the bacterial tyrosine kinase Wzc reduces polyadenylation activity, thereby stabilizing multiple mRNAs of stress-response genes. The authors identified tyrosine 202 of PAPI as critical target of phosphorylation. Consistent with such a pathway, wzc null mutant cells or cells expressing PAPI Y202F exhibit elevated sensitivity to stress.

General: The findings are interesting and novel and generally well presented in a clearly written manuscript with sound reasoning. Some presumably technical issues compromise the conclusions as specified below. These issues should be addressed to substantiate the proposed pathway.

Major points:

1. The detection of tyrosine phosphorylation relies entirely on experiments with exogenously expressed or recombinant proteins, and is not shown on endogenous protein, which is a limitation. This should be clearly indicated in the discussion.

Moreover, the experiments in Figs. 3 and 4 showing the tyrosine phosphorylation of PAPI are in my opinion not fully conclusive. The authors show in part relatively weak phosphorylation signals either as pTyr blots or as radioactive bands, which do not exactly match the bands shown as control blots or Coomassie bands, respectively (Fig. 3C, I, J, Fig. 4 E, F, Fig. 3K, Fig. 4G). This casts doubts that the phosphorylation is indeed on PAPI. For the pTyr blots, reblots should be shown so that identity of the bands can be concluded. The shown blots do not reveal identity, presumably because different membranes were probed. For the recombinant proteins, staining of the gels, which were subsequently subjected to phosphoimaging should be shown. In the present state it appears possible that the phosphorylation signals originate from contaminating or associating proteins. What is the origin of the double bands in Fig. 4E and F?

Ideal should be identification of the phosphorylation of Y202 by mass spectrometry. Given that the authors use as much as 100 μ g recombinant protein for in vitro phosphorylation (line 659), this may be possible and would greatly improve the study.

2. Identification of Wzc as critical kinase relies in part on Fig. 4 F, G. The signals in these images are rather weak making conclusions difficult. In particular for Fig. 4G an experiment with stronger signals appears mandatory to allow conclusions. The authors may consider to somewhat reduce the concentration of unlabeled ATP (currently 50 μ M) in their reactions to increase signal intensity.

3. The drop in polyadenylation activity of PAPI Y202F shown in Fig. S4A-B is a key finding, albeit this experiment is of lesser

quality than other experiments of similar type in this manuscript. The result should be moved to the main manuscript and integrated into Fig. 3, preferentially with a technically better experiment. In Fig. 4A the effect of the Y202F mutation on polyadenylation is shown by a RACE experiment. However, to make the point the results for Y-3 and Y-4 mutation should also be shown for control conditions.

4. The proposed mechanism would demand that the activity of Wzc on PAPI should be regulated by stress. While asking for such data appears too much, the issue should at least be discussed.

Minor points:

1. The authors state for Fig. 1 that 30% of regulated genes correspond to stress response. How does this compare to the percentage of stress response genes (by same criteria) in all genes? This comparison would show if there is indeed an enrichment.
2. Fig. 2A - the legend indicates analysis of the *uspC* gene - but this is not shown in the table, please correct.
3. Fig. S4E: Why is the image cut at the bottom, removing part of the area, which would show colonies for MG-wzc bacteria? Please, show complete image for this or another experiment.
4. Fig. 5: The description of D-G I found quite difficult to understand, please re-write (lanes 381-389).
5. There are small grammar issues, the text would benefit by an edit with help of a native speaker. For example in the Abstract are some small issues: line 44 - "the mechanism of the regulation ..(?)", line 47 would spell better "In contrast, PAPI expression..", line 48 "How does PAPI induce..", line 51 ".. tyrosine phosphorylation of PAPI that is strongly enhanced..".

(Unless otherwise indicated, page and figure numbers in parenthesis refer to the revised manuscript. The comments from the reviewers are presented in italics)

***Reviewer 1:** The conclusion that stress-induced inactivation is due to Tyr phosphorylation and not Ser/Thr phosphorylation of PAPI was based on blotting with anti-pSer and anti-pTyr Abs. However, anti-pSer Abs are known to be notoriously poor, and recognize only a subset of pSer phosphorylation sites. In other words, the data in Figure 3A/B do not rule out that Ser/Thr phosphorylation plays a role in the negative regulation of PAPI activity. Since the authors used ³²P-labeling to demonstrate increased phosphorylation of His-PAPI by lysates from stressed cells, they are in a position to carry out a direct phosphoamino acid analysis on ³²P-labeled PAPI to determine whether phospho-His-PAPI from stress-treated cells contains pSer/pThr in addition to pTyr.*

- As suggested, we have now carried out direct phosphoamino acid analysis of ³²P-labelled phospho-PAPI using one directional Thin Layer Chromatography (TLC). In this experiment, His-PAPI was phosphorylated in vitro in the presence of ³²P-radiolabelled γ ATP using stress treated cell lysate. The phospho-PAPI was then acid hydrolysed and analysed on a silica TLC plate along with unlabelled phosphoserine and phosphotyrosine amino acids as standards. We detected phosphorylated tyrosine but not serine residue from the acid hydrolysis of phospho-PAPI (Figure 3B). Consistent with our earlier experiments, there was increase in the level of phosphotyrosine obtained from acid hydrolysis of PAPI phosphorylated with stress-treated cell lysate compared to the untreated cell lysate (Figure 3E). There was a loss of the phosphotyrosine observed from the phospho-PAPI acid hydrolysis on Y202F mutation but not on other phosphomutations employed (Y60F, Y169F, Y170F, Y341F) (Figure 3F). Similarly, we also did not detect phosphotyrosine from acid hydrolysed PAPI after phosphorylation with lysates from *wzc*-null or kinase enfeebled (K540R) mutant cells (Figure 4H). Together, these results demonstrate that PAPI is phosphorylated at the tyrosine residue under stress treatment. (page 8;para 2, page 9; para1, page 11;para1)

To further confirm the Y202 PAPI phosphorylation, we generated an antibody specific to PAPI-Y202 using a phospho-PAPI peptide (IRLIGNPETR-Y(p)REDPVRMLR). The purified antibody detected the endogenous phosphorylated PAPI (referred to as Y202-phospho PAPI and the antibody as Y202-phospho antibody) and phosphorylation on exogenously expressed PAPI in both stress-treated and untreated MG1655 cells (Figure 3M-O). Competition with corresponding phospho and non-phospho peptides demonstrated specificity of the Y202-phospho antibody in Western blot analysis (Figure S3J). The Y202-phospho peptide specifically competed with the Y202-phospho antibody but not by the non-phospho PAPI peptide (Figure S3J). Additionally, specificity of the antibody was further confirmed in a Western blot analysis using cell lysates expressing Y202F-PAPI mutant protein, and in lysates from *pcnB* null mutant cells (Figure 3P). Consistent

with our earlier results using general phosphotyrosine antibody, Western blot analysis with our Y202-phospho antibody showed increased endogenous PAPI phosphorylation in the presence of stress treatment (Figure 3M). Similar inductions were also observed from ectopically expressed FLAG-PAPI in the cell in the presence and absence of different stress treatment (Figure 3N). Using the same Y202-phospho antibody, we have also demonstrated the loss of PAPI phosphorylation in the presence of kinase *wzc*-null and kinase enfeebled *wzc*-K540R mutations in the cell (Figure 4D). (page11;para 2, page 14;para1)

2: The authors provide no direct evidence that Tyr202 in PAPI is phosphorylated in response to stress - the effect of the Y202F mutation could be indirect. The generation and use of a pY202 PAPI-specific antiserum would be one way to establish this. Alternatively, they should carry out direct MS-based phosphoproteomic analysis on either Flag-PAPI from stress-treated cells or phospho-His-PAPI incubated in stress-treated cell lysates to identify the stress-induced sites of phosphorylation, i.e., both pTyr and pSer/pThr sites, and demonstrate phosphorylation of Y202. As it stands, while the Y202F mutation results indicate that Y202 is needed for reduction in enzymatic activity, Y202 phosphorylation might not be sufficient.

- As suggested by the reviewer, we have now generated Y202 phospho-PAPI specific antibody (Y202-phospho antibody) to confirm the endogenous phosphorylation of PAPI. As described above, specificity and the efficacy of the purified antibody was tested using competition with PAPI-phospho and non-phospho peptides, and from cells with or without Y202 PAPI mutation (Figure S3J). By Western blot analysis using the Y202-phospho antibody, we demonstrated an induction of endogenous PAPI tyrosine phosphorylation in the cell on treatment with multiple stress conditions (Figure 3M). Similar phosphorylation was also confirmed using ectopically expressed FLAG-PAPI in the cell in the presence of different stress treatments (Figure 3N). Moreover, the stress primed tyrosine phosphorylation was lost on Y202F mutation but not on other phosphomutations (Y60F, Y169F, Y170F, Y341F) on PAPI (Figure 3P). The same tyrosine phosphorylation was not detected in the cell harboring bacterial tyrosine kinase *wzc*-null or kinase enfeebled *wzc*-K540R mutations but not other tyrosine kinase mutations tested (*bipA*, *etk*, or *ydiB*) (Figure 4D). Together with our earlier results, these results demonstrate that PAPI is phosphorylated at the tyrosine residue that is induced during stress response. (page11;para 2, page 14;para1) (also see response to comment 1, above)

3. As evidence that phosphorylation of PAPI negatively regulates its enzymatic activity, the authors showed that treatment of PAPI isolated from stress-treated cells with λ phosphatase led to increased polyadenylation activity. They concluded that this increase was due to dephosphorylation of pTyr in PAPI, which resulted in enzymatic reactivation. However, they did not show that λ phosphatase treatment actually caused loss of the pTyr signal, which is essential for them to demonstrate. They also need to include a control in which the λ phosphatase treatment of PAPI is carried out in the presence of a phosphatase inhibitor. Most importantly, λ phosphatase is not specific for pTyr, but can also dephosphorylate pSer/pThr, and so the fact that λ phosphatase treatment increased PAPI polyadenylation activity does not demonstrate that the stress-induced decrease in PAPI activity was due solely to Tyr phosphorylation. To establish this, the authors need to redo this experiment treating PAPI from stress-treated cells with a pTyr-specific phosphatase, such as recombinant

YopH, showing that this causes a loss of pTyr in PAPI and also leads to enzymatic reactivation.

- As suggested, we have now carried out new in vitro polyadenylation assay using purified PAPI from stress-primed cells after treatment with λ phosphatase in the presence and absence of a lambda phosphatase inhibitor (sodium vanadate) (Figure 3I, H). We observed reduction of PAPI activity from the stress treated cell lysate that was ameliorated after dephosphorylation by λ -phosphatase. However, addition sodium vanadate inhibited λ -phosphatase activity on PAPI that resulted in the loss of PAPI polyadenylation activity similar to stress treated cell lysates. However, there was no effect of addition of sodium vanadate on the PAPI activity in the control experiment in absence of λ -phosphatase (Figure 3I, H). (page 10; para 1)

Further, to assess the specific tyrosine phosphatase activity, purified PAPI from stress primed cell lysate was treated with a phosphotyrosine specific bacterial phosphatase (*YopH* from *Yersinia pestis*) as suggested. Similar to λ -phosphatase, dephosphorylation with *YopH* augmented the PAPI activity that was reduced on stress treatment. Similar to λ -phosphatase, *YopH* treatment did not show any significant increase in the PAPI activity in the presence of sodium vanadate (Figure 3I, H). Together, these experiments demonstrate that phosphorylation of PAPI negatively regulates PAPI enzymatic activity. Western analysis of the phosphorylation pattern of PAPI in the presence and absence of λ -phosphatase, and *YopH* with its inhibitor treatment is shown in Figure S3G. (page 10; para 1)

4. For stress-induced phosphorylation of Y202 to efficiently inhibit PAPI polyadenylation activity, the stoichiometry of Y202 phosphorylation would have to be high. Did the authors estimate the stoichiometry of PAPI Tyr phosphorylation, either in vivo or in vitro? For instance, if a higher resolution gel is run, is the mobility of PAPI shifted upon phosphorylation and restored when the sample is treated with λ phosphatase?

- In our Western analysis using normal 8% and 10% acrylamide gels (or even higher 15% gels), we did not observe an altered mobility or supershift of PAPI protein from increased phosphorylation. We could not use above 15% due to technical difficulties in the Western transfer of higher molecular weight proteins such as PAPI. However, we clearly detected and validated an induction of phosphotyrosine on stress treatment using both Y202-phospho antibody and direct phosphoamino acid analysis by acid hydrolysis. Nevertheless, we observed decrease in phospho PAPI level (in the purified FLAG-PAPI) on λ -phosphatase or *YopH* treatment that was inactivated in the presence of sodium vanadate while FLAG-PAPI remained unchanged (Figure S3G). However, we cannot infer on the stoichiometry of PAPI phosphorylated fraction from these gels as they are in different blots. At present, the stoichiometry of Y202-phospho versus non-phospho PAPI fractions in the cell is still unclear. (page 10, para 9)

5. Assuming that the authors' conclusions are correct, there are no insights into how a phosphate linked to Tyr202 would inhibit PAPI catalytic activity - is there a structure of the PAPI catalytic domain or of a related polymerase that could be used to model how Tyr202 phosphorylation might affect poly(A) polymerase activity? D212 and R215 are important catalytic residues - could the phosphate on Y202 interfere with one or both of these residues, for instance, by charge repulsion for

D212 or interaction with R215? Is the OH of Y202 predicted to be accessible for phosphorylation by Wzc in the active site?

- To understand how Y202 phosphorylation affects PAPI polyadenylation activity, we analysed the Y202 phosphorylation-induced structural changes using MD simulations spanning 300 ns for PAPI and Y202 phospho-PAPI. In simulations initiated from PAPI, the root mean square deviation (RMSD) of protein backbone of PAPI was significantly increased in Y202 phosphorylation indicating an overall structural alteration (Figure 4A, top panel). Moreover, the Y202 phosphorylation perturbed several interactions including formation of new H-bonds in the MD simulated structure (Figure S4F). For example, upon phosphorylation, Y202 residue forms a new hydrogen bond with Threonine 227 (T227) with a bond length of ~ 2.7 Å (Figure S4E, and 4A lower panel) that could affect the ATP binding by repositioning of amino acids in the pocket. Strikingly, we observed loss of six out of seven hydrogen bonds that were known to stabilise the ATP-PAPI interaction after Y202 phosphorylation (Toh et al. 2011) (Figure 4B). Among the critical intramolecular interactions, the distance between D205 and R211 (that are known to affect PAPI activity) was increased from 3.7 Å to 8.3 Å inducing an overall reduction in the hydrogen bond in the ATP binding pocket throughout the simulation (Figure S4F). As a result, as simulation progresses, ATP leaves the binding pocket in the phospho Y202 PAPI unlike in the non-phospho PAPI (Figure 4A-B, S4F). Y202 site is also exposed in a way to be accessible for the kinase.

Together, these results indicate that Y202 phosphorylation induces allosteric changes in the ATP binding pocket of PAPI that potentially reduces its polyadenylation activity (page 12; para 2, page 13; para 1).

6. The evidence that Wzc BY tyrosine kinase directly phosphorylates PAPI at Y202 looks reasonable, but Wzc has not been shown convincingly to phosphorylate any protein other than itself, where autophosphorylation of its C-terminus is involved in polysaccharide biosynthesis. Is there a kinase-dead point mutation of Wzc that could be used as another control? If Wzc is the Tyr kinase responsible for phosphorylating PAPI upon stress treatment, how is it activated by stress - for instance, does Wzc exhibit increased autophosphorylation upon stress treatment. Or is Wzc constitutively active, with the level of pY202 in PAPI being controlled by a stress-regulated phosphatase? What is the pTyr phosphatase that dephosphorylates pY202 PAPI and restores its activity, when the stress response is resolved? In this regard, the authors do not show how rapidly the pTyr signal in PAPI is lost when the stressor is removed.

- As reviewer rightly pointed out, Wzc auto phosphorylates itself through its autokinase activity in the presence of ATP. Similarly, Wzc is also known to phosphorylate few other target proteins such as UDP-glucose dehydrogenase (ugd) and phage integrase (int)(Grangeasse et al. 2002; Grangeasse et al. 2003; Kolot et al. 2008). To assess if inactivation of Wzc kinase activity will abolish PAPI phosphorylation, we employed Wzc-K540R point mutation (kinase enfeebled) that is reported to abrogate its autophosphorylation and phosphorylation on target Ugd protein (Grangeasse et al. 2002; Grangeasse et al. 2003). Using a TLC-based phosphoaminoacid analysis after in vitro phosphorylation of PAPI using purified Wzc, we showed that Wzc-K540R failed to phosphorylate PAPI while it was phosphorylated by wild type Wzc (Figure 4H). Similarly, in a Western blot analysis

using Y202-phospho antibody, we did not detect PAPI tyrosine phosphorylation in the presence of Wzc-K540R or *wzc*-null mutation (Figure 4F). We further confirmed these findings in a polyadenylation assay of PAPI with prior in vitro phosphorylation using wild type Wzc or Wzc-K540R mutant protein (Figure 4J). We observed reduced polyadenylation activity of PAPI after phosphorylation with wild type Wzc (Figure 4J). Together, these results demonstrate that Wzc kinase phosphorylates PAPI that reduces PAPI polyadenylation activity. (page 15; para2)

- To further address the question as how Wzc is regulated during stress that would act as upstream signal for PAPI-mediated stress response gene expression, we analysed both *wzc* expression and Wzc autophosphorylation upon stress treatment. Analysis of promoter sequence for *wzc* revealed that transcription of *wzc* operon is controlled by sigma factor RpoE (σ_{24}) for *wzc* promoter 1 (*wzcp1*) in addition to the general sigma factor RpoD for *wzc* promoter 2 (*wzcp2*) (sequence of the *wzc* operon is shown in Figure S5D). RpoE is known to regulate transcription of genes involved in response to growth in high temperature, hyper osmotic shock or protein unfolded response. Earlier, it was also shown that *wzc* expression is induced during osmotic stress. Additionally, *wzc* expression is regulated by Rcs phosphorelay system, a well-established regulator of envelope stress. The transcriptional activator RcsAB binds upstream of *wzcp1* promoter and enhances transcription of *wzc* operon. Therefore, we assessed the expression levels of *wzc* transcript in presence and absence of different stresses (Figure 4M). We observed increased *wzc* transcript level when treated with different stresses (osmotic shock, acid shock or cold shock). The control stress regulated transcript, *osmY* was also equally induced under stress treatment but not the non-stress related transcript *dxs*. However, we did not observe any significant difference in the Wzc auto phosphorylation in the presence and absence of stress treatment (Figure S5E). These results indicate that *wzc* expression is controlled through transcription under stress that further regulates PAPI activity during stress response. (page 16; para 1)

Minor Points:

1. Figure 3A/B: The level of pTyr in PAPI was significant in these samples, but it is not stated in the figure legend whether the cells from which these samples were obtained had been stress treated.

- Modified as suggested

2. Figure 3G: There seems to be no description of how the bacterial cells were lysed and the cell lysates prepared (e.g. what was the buffer and is this compatible with the phosphorylation reactions carried out with the cell lysates?) This may be described elsewhere, but detailed methods need to be provided). Likewise, there is no description of how the lysate phosphorylation assays were carried, i.e. what was the protein concentration, the final concentration of added ATP, incubation time and temperature, etc.

- All reactions and lysate preparations were carried out in Tris-Cl buffer. To maintain the compatibility of the buffers in all subsequent reactions, where in vitro

kinase assay was performed prior to λ -phosphatase treatment or polyadenylation assay, buffers were supplemented with required salts to maintain similar composition. The detailed methods, buffer compositions and reaction conditions are now described in the Materials and Methods. (page 26; para 1)

3. *Figure 3H: The M1-3 boxes in the schematic need to be defined in the legend?*

- Modified as suggested.

4. *Page 9: The NetPhos prediction algorithm was developed to predict phosphorylation sites in eukaryotic proteins, and there is no reason to believe that this algorithm will be successful in predicting (Tyr) phosphorylation sites in bacterial proteins, where protein phosphorylation is mediated by totally different types of (Tyr) protein kinase to those used in eukaryotes.*

- As rightly pointed out by the reviewer, Net.Phos3.1 prediction algorithm was developed to predict phosphorylation sites in eukaryotic proteins. Net.Phos.Bac, a prediction tool for identifying residues in bacterial protein did not predict tyrosine phosphorylation and it identifies only serine/threonine phosphorylation sites. Our experimental data indicated that PAPI is phosphorylated at the tyrosine residue during stress treatment. Therefore, we limited our search to putative tyrosine phosphorylation residues. While we were unable to find suitable in silico bacterial phosphorylation prediction tool for PAPI, Net.Phos3.1 predicted at least 5 phosphorylation sites above the threshold. Further, with a series of experiments including biochemical, genetic and physiological, we demonstrated that PAPI is phosphorylated at the Y202 residue that is induced during stress response. (page 10; para 2)

5. *Figure 3K: The fact that there was no background in the Y3 lane, compared to that observed in WT, Y1, Y2 and Y4 lanes is odd, and needs to be explained.*

Moreover, based on the shapes of the His-PAPI bands in the lower panel, it does not look as though these are the same as those that were phosphorylated. Exactly how this experiment was carried out needs to be explained in more detail.

- This gel has been removed. As suggested, we have now repeated the experiment and detected the Y202-phosphorylation using direct phosphoamino acid analysis with one directional TLC. We consistently show p32 incorporation in on His-PAPI and other phosphomutants in the in vitro phosphorylation reaction using stress primed cell lysates that was not detected in the presence of Y202F mutation (Figure 3F). (page 11; para 2)

In our earlier in vitro kinase experiments (Fig 3K in our original manuscript), we use ~20 μ g of total cell lysate protein in our in vitro kinase reaction and incubated with ~10 μ g of substrate PAPI protein. In the coomassie staining of the gel, proteins from this cell lysate mask the substrate PAPI and makes it difficult to visualise the substrate band (coomassie stain gel of in vitro kinase reaction and the substrate protein along is shown in rebuttal Figure 1). Hence, we could not stain and expose the same gel for the same reaction. Therefore, for all in vitro kinase experiments, we used duplicate gels, one for radioactive exposure and another for substrate protein staining without the cell lysates. Nevertheless, the two reactions and gels were analysed under

exact same experimental and technical conditions (also see response for comment 2, reviewer 2).

6. *Figure S3A: As was the case for Figure 3K, it does not look as though the His-PAPI bands in the lower panel are the same as those that were phosphorylated/blotted with anti-pTyr Ab in the upper panel. Why is the upper panel labeled 32P, when the legend says that these samples were blotted with anti-pTyr Abs?*

- There was a typo in the labelling of Fig S3A. The upper band is a blot for phosphotyrosine (blotted with anti-phospho-tyrosine antibody) while the lower band is blot for His-PAPI with anti-His antibody. We have now corrected the labelling (Figure S3C). (Supplementary data page 3, para 1)

Moreover, because of the same sizes of both His-PAP and phospho-Tyr PAPI on the blot, we could not use the same blot for detecting both proteins. Moreover, stripping the same blot and reblotting did not work for most of our antibodies. Therefore, we use duplicate gels analysed under the same experimental and technical conditions and blotted using phospho- and non-phospho PAPI antibodies. (also see the response for comment 5, reviewer 2).

7. *Figure S3C: This experiment should include a λ phosphatase control to show that this restores the polyadenylation activity of His-PAPI exposed to the lysate from stress-treated cells, and also include a control in which the λ phosphatase is added to the unphosphorylated His-PAPI protein, and another where a λ phosphatase inhibitor is added.*

- Figure S3C did not have λ -phosphatase treatment. This has now been clarified in the in the revised figure labelling. Similar polyadenylation assays in the presence and absence of λ -phosphatase and its inhibitors is now shown in Figure 3I. (page 9; para 2) (also see response to comments 3, above).

8. *Figure S3D: The right-hand lane where the activity of the dephosphorylated Flag-PAPI was measured has a very strong band that was not obvious in the other assay lanes - do the authors have an explanation? Why did the authors not quantify these three lanes like they did those in panel C? It is essential that the authors repeat this experiment to show that the λ phosphatase treatment completely removed the Flag-PAPI pTyr signal, and also include a control in which the λ phosphatase is added to the unphosphorylated Flag-PAPI protein.*

- We have now repeated the polyadenylation assay with additional conditions as suggested. In the new experiment, PAPI was in vitro phosphorylated with stress cell lysates in the presence and absence of λ -phosphatase or specific bacterial tyrosine phosphatase YopH and its inhibitors (a general phosphatase inhibitor, sodium vanadate) (Figure 3I). The quantification of the bands are shown in Figure S3H (page 9; para 2) (also see response to comments 3, above)

9. *Figure 4F and G: These Wzc phosphorylation data are not totally convincing. In panel F, the pTyr-positive bands in the upper panel are rather faint, and need quantifying, and again do not seem to correspond to the His-positive bands in the*

lower panel. In panel G, the fact that the Y4 and Y3 samples are not in order is confusing, and the very high background in the Y4 lane needs explaining. Also, these bands need quantifying. Did the authors measure the polyadenylation activity of PAPI isolated from *wzc*-null cells compared to WT cells? Did they show that Flag-Wzc phosphorylation of PAPI reduced its polyadenylation activity?

- To address the reviewers concern, we have now repeated the in vitro phosphorylation assay and employed Y202-phospho antibody to detect the PAPI tyrosine phosphorylation. We showed a loss of PAPI tyrosine phosphorylation in the presence of Y202F mutation but not on other phosphomutations (Figure 4E, G). (page 14; para 1) (also see response comments 1 and 2, above)

As suggested, we have also incorporated an in vitro polyadenylation assay of PAPI after in vitro phosphorylation with stress treated cell lysates from normal and *wzc* mutant cells (Figure 4I). We observed reduction in PAPI activity after in vitro phosphorylation with stress treated cell lysates that was ameliorated by *wzc* mutation. Similarly, in an in vitro polyadenylation assay of PAPI after phosphorylation with purified Wzc showed reduction in the polyadenylation activity (Figure 4J). (page 15; para 1)

10. *pY202 does not seem to have been identified in published E. coli pTyr phosphoproteomic databases, but this should be checked more thoroughly (the pY202 tryptic peptide may be too small to be identified by MS).*

- As rightly pointed out by the reviewer, published phosphotyrosine phosphoproteomic data and other database did not reveal Y202 phospho peptide of PAPI. This is a novel phosphorylation site that we detected in silico and confirmed subsequently with a series of genetic, biochemical and physiological experiments. (page 11, para 2)

Reviewer #2 (Comments to the Authors (Required)):

*Francis et al. propose a novel pathway of regulation of stress response in E.coli, downstream of the master stress regulator RpoS. According to their findings, stress-induced tyrosine phosphorylation of poly (A) polymerase (encoded by *pcnB*) through the bacterial tyrosine kinase Wzc reduces polyadenylation activity, thereby stabilizing multiple mRNAs of stress-response genes. The authors identified tyrosine 202 of PAPI as critical target of phosphorylation. Consistent with such a pathway, *wzc* null mutant cells or cells expressing PAPI Y202F exhibit elevated sensitivity to stress.*

General: The findings are interesting and novel and generally well presented in a clearly written manuscript with sound reasoning. Some presumably technical issues compromise the conclusions as specified below. These issues should be addressed to substantiate the proposed pathway.

- We thank the reviewer for critically assessing our work and we have now addressed concerns raised in our original manuscript with new experiments and/or explanations as appropriate.

Major points:

1. The detection of tyrosine phosphorylation relies entirely on experiments with exogenously expressed or recombinant proteins, and is not shown on endogenous protein, which is a limitation. This should be clearly indicated in the discussion.

- To address the reviewers concern and to confirm the Y202 PAPI phosphorylation, we generated an antibody specific to PAPI-Y202 using a phospho-PAPI peptide (IRLIGNPETR-Y(p)REDPVRMLR). The purified antibody detected the endogenous phosphorylated PAPI (referred to as Y202-phospho PAPI and the antibody as Y202-phospho antibody) and phosphorylation on exogenously expressed PAPI in both stress-treated and untreated MG1655 cells. Competition with corresponding phospho and non-phospho peptides demonstrated specificity of the Y202-phospho antibody in Western blot analysis (Figure S3J). The Y202-phospho peptide specifically competed with the Y202-phospho antibody but not by the non-phospho PAPI peptide (Figure S3J). Additionally, specificity of the antibody was further confirmed in a Western blot analysis using cell lysates expressing Y202F-PAPI mutant protein, and in lysates from *pcnB* null mutant cells (Figure 3P). Consistent with our earlier results using general phosphotyrosine antibody, Western blot analysis with our Y202-phospho antibody showed increased endogenous PAPI phosphorylation in the presence of stress treatment (Figure 3P). Similar inductions were also observed from ectopically expressed FLAG-PAPI in the cell in the presence and absence of different stress treatment (Figure 3N,O). Using the same Y202-phospho antibody, we have also demonstrated the loss of PAPI phosphorylation in the presence of kinase *wzc*-null and kinase enfeeblled *wzc*-K540R mutations in the cell (Figure 4F). (page 8; para 2, page 9; para 1, page 11; para 1).

(also see response to comments 1 and 2, Reviewer 1)

Moreover, the experiments in Figs. 3 and 4 showing the tyrosine phosphorylation of PAPI are in my opinion not fully conclusive. The authors show in part relatively weak phosphorylation signals either as pTyr blots or as radioactive bands, which do not exactly match the bands shown as control blots or Coomassie bands, respectively (Fig. 3C, I, J, Fig. 4 E, F, Fig. 3K, Fig. 4G). For the recombinant proteins, staining of the gels, which were subsequently subjected to phosphor imaging should be shown. In the present state it appears possible that the phosphorylation signals originate from contaminating or associating proteins.

- Figure 3C, 3I, 3J, 4E, 4F are Western blots images and Figure 3K, 4G are in vitro kinase assays. First for the kinase assays, Figure 3K and 4G (original manuscript) has now been removed and replaced with a new gels of in vitro kinase assay detected with direct phosphoaminoacid analysis (Figure 3F) and specific Y202-phospho antibody after cold in vitro phosphorylation reaction (Figure 4G). Mutational analysis of Y202 and also the Wzc kinase that phosphorylates PAPI demonstrates that the phospho band is specific (Figure 3P, 4D). Consistently, while the signals are not very sharp, we clearly see radioactive incorporations in the new in vitro kinase assays that are lost on phospho mutation. (page 14; para 1)

- In our earlier in vitro kinase experiments (Fig 3K and 4G in our original manuscript), we use ~20 µg of total cell lysate protein in our in vitro kinase reaction

and incubated with $\sim 2 \mu\text{g}$ of substrate His-PAPI protein. In the coomassie stain of the gel, proteins from this cell lysate mask the substrate PAPI and make it difficult to visualise the substrate band (representative coomassie stain gel of in vitro kinase reaction and the substrate protein alone is shown in Rebuttal Figure 1, below). As shown in the figure, in the coomassie stained gel, substrate PAPI is undetectable in the presence of cell lysate. Hence, for all in vitro kinase experiments, we used duplicate gels, one for radioactive exposure and another for substrate protein staining without the cell lysates. Nevertheless, the two reactions and gels analysed under exact same experimental and technical conditions. Moreover, we have also repeated the experiments and detected using more reliable and specific Y202-phospho antibody by Western analysis. (page 11, para 1) (see response to comments 5 and 6, reviewer 1)

Rebuttal Figure 1: Representative coomassie images of substrate PAPI and cell lysates in the in vitro kinase assay as in Figure 3D. (A) Purified PAPI analysed separately in the exact amount as used in the in vitro kinase assay in Figure 3D (B) Coomassie staining of equal loading of cell lysates used as in Figure 3D. A purified PAPI run along with the gel for comparison is indicated. (C) Coomassie stain of in vitro kinase reaction mix (containing both substrate purified PAPI and cell lysate used in Figure 3D. A separate lane of purified PAPI alone is indicated for comparison.

For the pTyr blots, reblots should be shown so that identity of the bands can be concluded. The shown blots do not reveal identity, presumably because different membranes were probed.

- Figure 3C, 3I, 3J, 4E, 4F are blots of phospho PAPI with general phospho-tyrosine antibody. To address the reviewer's concern and confirm the Y202 phosphorylation detected in these gels, we have now repeated the Western analysis using phospho-Y202 specific antibody that detects specifically the PAPI-Y202 phospho protein (Figure 3M, 3P, 4D). Consistent to our earlier results, we show induced Y202 phosphorylation that is lost on the phosphomutation or the kinase mutation (Figure 3M, 3P, 4D). Moreover, we are also able to detect endogenous PAPI phosphorylation with our new phospho-Y202 specific antibody (Figure 3M). With the new incorporations, we can conclude reliably that PAPI is phosphorylated at the Y202 that is induced under stress. (page 11;para 1)

As in the case of in vitro kinase assays, we used duplicate membranes to probe for PAPI and phospho-PAPI proteins. Because of the same sizes of both His-PAP/FLAG-PAPI and phospho-Tyr PAPI on the membrane, we could not use the same membrane for blotting both proteins. Moreover, stripping the same membrane and reblotting did not work for most of our antibodies. Therefore, we use duplicate gels analysed under the same experimental and technical conditions and blot with both antibodies separately. (also see response to comment 5,6, reviewer 1).

Ideal should be identification of the phosphorylation of Y202 by mass spectrometry. Given that the authors use as much as 100 µg recombinant protein for in vitro phosphorylation (line 659), this may be possible and would greatly improve the study.

- Mass spectrometry analysis failed to detect the peptide region of the PAPI Y202 phospho site in our experiment. As a result we could not detect the phospho Y202. As suggested by the reviewer and also by the reviewer 1, we have now generated a Y202 phospho specific PAPI antibody and demonstrated PAPI endogenous and direct phosphorylation in the cell (Figure 3M-P, 4D,F). In addition, we have also carried out direct phosphoamino acid analysis using one directional TLC to confirm the tyrosine phosphorylation of PAPI as suggested by the reviewer 1 (Figure 3B,E,F, 4H). Together, these results along with our earlier data clearly demonstrate Y202 phosphorylation of PAPI that is induced during stress response. (page 11, para 1) (also see response to comments 1,2, reviewer 1 and comments 1 above)

2. Identification of Wzc as critical kinase relies in part on Fig. 4 F, G. The signals in these images are rather weak making conclusions difficult. In particular for Fig. 4G an experiment with stronger signals appears mandatory to allow conclusions. The authors may consider to somewhat reduce the concentration of unlabeled ATP (currently 50 µM) in their reactions to increase signal intensity.

- As suggested, Figure 4F and G has now been repeated and probed with Y202-phospho specific antibody (Figure 4E,G). (page 14, para 1)

3. The drop in polyadenylation activity of PAPI Y202F shown in Fig. S4A-B is a key finding, albeit this experiment is of lesser quality than other experiments of similar type in this manuscript. The result should be moved to the main manuscript and integrated into Fig. 3, preferentially with a technically better experiment. In Fig. 4A the effect of the Y202F mutation on polyadenylation is shown by a RACE experiment. However, to make the point the results for Y-3 and Y-4 mutation should also be shown for control conditions.

- As reviewer rightly pointed out, the reduction in the PAPI activity from Y202 mutation is an important data. In addition, we have also included new polyadenylation assays of PAPI after in vitro phosphorylation with purified Wzc kinase (Figure 4J) and in the presence of lambda phosphatase with and without specific inhibitors (Figure 3I). This now clearly shows effect of phosphorylation on PAPI activity. However, due to space constraint in the Figure 3, the in vitro polyadenylation assay has been retained in the Supplementary Figure S4 A,B. (page 12; para 1)

Our 3'-RACE shows loss of polyadenylation of *osmY* transcript under stress that is rescued by the P-Y202 mutation but not by the wild type or Y341F PAPI mutation. This shows the loss of polyadenylation under stress that is rescue with Y202F but not WT or other Y341F mutant.

4. The proposed mechanism would demand that the activity of Wzc on PAPI should be regulated by stress. While asking for such data appears too much, the issue should at least be discussed.

- To address the question as how Wzc is regulated during stress that would act as upstream signal for PAPI-mediated stress response gene expression, we analysed both *wzc* expression and Wzc autophosphorylation upon stress treatment. Analysis of promoter for *wzc* revealed that transcription of *wzc* operon is controlled by sigma factor RpoE (σ_{24}) for *wzc* promoter 1 (*wzcp1*) in addition to the general sigma factor RpoD for *wzc* promoter 2 (*wzcp2*) (sequence of the *wzc* operon is shown in Figure S5D). RpoE is known to regulate transcription of genes involved in response to growth in high temperature, hyper osmotic shock or protein unfolded response. Earlier, it was also shown that *wzc* expression is induced during osmotic stress. Additionally, *wzc* expression is regulated by Rcs phosphorelay system, a well-established regulator of envelope stress. The transcriptional activator RcsAB binds upstream of *wzcp1* promoter and enhances transcription of *wzc* operon. Therefore, we assessed the expression levels of *wzc* transcript in presence and absence of different stresses (Figure 4M). We observed increased *wzc* transcript level when treated with different stresses (osmotic shock, acid shock or cold shock). The control stress regulated transcript, *osmY* was also equally induced under stress treatment but not the non-stress related transcript *dxs*. However, we did not observe any significant difference in the Wzc auto phosphorylation in the presence and absence of stress treatment (Figure S5E). These results indicate that *wzc* expression is controlled through transcription under stress that further regulates PAPI activity during stress response. (page 15; para 2, page 16; para 1)

Minor points:

1. The authors state for Fig. 1 that 30% of regulated genes correspond to stress response. How does this compare to the percentage of stress response genes (by same criteria) in all genes? This comparison would show if there is indeed enrichment.

- Around 30% of the genes from the up regulated (PAPI targets) on *pcnB* mutation correspond to stress response genes (Figure 1C). However, we analysed among the total genes altered/detected on *pcnB* mutation, we observed ~30.5% of stress response genes. Interestingly, if we consider the total stress response related genes detected on *pcnB* mutation, majority of them are detected among the upregulated genes (>90%). This clearly shows the enrichment of stress related genes among the *pcnB* targets.

2. Fig. 2A - the legend indicates analysis of the *uspC* gene - but this is not shown in the table, please correct.

- We have modified in the revised draft. (page 40, para 2)

3. Fig. S4E: Why is the image cut at the bottom, removing part of the area, which would show colonies for MG-wzc bacteria? Please, show complete image for this or another experiment.

- We have now modified the image in the revised draft. (Figure S5C, page 15; para 2)

4. Fig. 5: The description of D-G I found quite difficult to understand, please re-write (lanes 381-389).

- We have now modified in the revised draft. (page 45, para 2)

5. There are small grammar issues, the text would benefit by an edit with help of a native speaker. For example in the Abstract are some small issues: line 44 - "the mechanism of the regulation ..(?), line 47 would spell better "In contrast, PAPI expression..", line 48 "How does PAPI induce..", line 51 ".. tyrosine phosphorylation of PAPI that is strongly enhanced..".

- We have now revised the manuscript for grammatical errors and language.

References:

- Grangeasse C, Doublet P, Cozzone AJ. 2002. Tyrosine phosphorylation of protein kinase Wzc from Escherichia coli K12 occurs through a two-step process. *J Biol Chem* **277**: 7127-7135.
- Grangeasse C, Obadia B, Mijakovic I, Deutscher J, Cozzone AJ, Doublet P. 2003. Autophosphorylation of the Escherichia coli protein kinase Wzc regulates tyrosine phosphorylation of Ugd, a UDP-glucose dehydrogenase. *J Biol Chem* **278**: 39323-39329.

- Kolot M, Gorovits R, Silberstein N, Fichtman B, Yagil E. 2008. Phosphorylation of the integrase protein of coliphage HK022. *Virology* **375**: 383-390.
- Toh Y, Takeshita D, Nagaike T, Numata T, Tomita K. 2011. Mechanism for the alteration of the substrate specificities of template-independent RNA polymerases. *Structure* **19**: 232-243.

November 4, 2022

RE: Life Science Alliance Manuscript #LSA-2021-01148-TR

Dr. Rakesh S. Laishram
Rajiv Gandhi Centre for Biotechnology
Cancer Research program
Thycaud Post, Poojappura
Trivandrum 695014
India

Dear Dr. Laishram,

Thank you for submitting your revised manuscript entitled "Tyrosine phosphorylation controlled poly(A)polymerase activity regulates stress response in bacteria". We would be happy to publish your paper in Life Science Alliance pending final revisions necessary to meet our formatting guidelines.

- please address the remaining Reviewer 1' comments
- please add ORCID ID for corresponding author-you should have received instructions on how to do so
- please include the file named Supplemental Text into your main Materials & Methods section
- please provide the Accession number for your deposited RNA seq data in Data Availability section

Figure Check:

- Figure S5 A and B: is the 2nd row a duplicate? The background in the column 2 (A) and the column 3 (B) looks the same. Please provide source data for these two panels

A. FINAL FILES:

B. MANUSCRIPT ORGANIZATION AND FORMATTING:

Sincerely,

Reviewer #1 (Comments to the Authors (Required)):

The authors have done an excellent job of addressing the reviewers' concerns, and in particular my request for better biochemical evidence that phosphorylation of Tyr202 in PAPI by Wzc is important for its stress induced inactivation. They have included a large number of new experiments to demonstrate that PAPI is phosphorylated on Tyr202 by Wzc under stress conditions and that this inhibits its activity, which normally polyadenylates stress response mRNAs leading to their destabilization. The one thing they were not able to do was to establish the stoichiometry of phosphorylation of Tyr202 in PAPI, which needs to be high for strong downregulation of PAPI polyadenylation activity under stress conditions - this could be determined by quantitative MS analysis of the pY202 and Y202 peptides from PAPI isolated from stressed and unstressed cells. However, it is not necessary for this experiment to be done before publication.

Points: 1. Figure 3P: These data are not very impressive, because the WT pY202 PAPI signal is so low.

2. Figure 4A/B: It is no unreasonable that a phosphate on Y202 will cause a conformational change in the ATP-binding pocket. However, while the structural simulation analysis suggests that Y202 phosphorylation might prevent ATP association, it does not prove this. In consequence, the authors should be careful about their conclusions, unless they carry out ATP binding studies on unphosphorylated and pY202 PAPI to demonstrate pY202 PAPI has decreased affinity for ATP.

3. Figures 3P/4G: From these two panels, it almost appears as though the Y314F PAPI mutation may promote Y202 phosphorylation - do the authors have any insights?

4. Figures 4M/S5E: The authors showed that Wzc RNA levels were increased by stress but apparently did not show that Wzc protein level is upregulated, which is the mechanism they are proposing for how PAPI Y202 phosphorylation is increased during stress. In Figure S5E, it appears that the same level of FLAG-Wzc protein was present in the stressed and unstressed cells, but its activity towards PAPI Y202 was strongly increased. Since Wzc phosphorylation would normally be regarded as a measure of its enzyme activity, it is somewhat surprising that Wzc autophosphorylation was not increased by stress when PAPI phosphorylation was strongly elevated. How do the authors explain this?

5. A model figure incorporating their findings/conclusions would help the reader.

Reviewer #2 (Comments to the Authors (Required)):

The authors have carefully addressed all issues raised in my previous review. By new experiments and by text amendments the manuscript has been greatly improved.

Notably, the authors generated a site-specific antibody against PAPI pTyr202, and key experiments with this antibody showed that tyrosine phosphorylation occurs as proposed before, and has a regulatory function. Moreover, the antibody enabled the authors to show tyrosine phosphorylation of endogenous PAPI. Thereby all previously remaining concerns about the identity of the phosphorylated protein were resolved.

All other points were also addressed by new experiments or improvements of the text. Therefore, I strongly recommend publication of this manuscript in its present form.

Thank you for submitting your revised manuscript entitled "Tyrosine phosphorylation controlled poly(A)polymerase activity regulates stress response in bacteria". We would be happy to publish your paper in Life Science Alliance pending final revisions necessary to meet our formatting guidelines. Along with points mentioned below, please tend to the following:

- please address the remaining Reviewer 1 comments

- Added as suggested

- please add ORCID ID for corresponding author-you should have received instructions on how to do so

- Added as suggested

- please include the file named Supplemental Text into your main Materials & Methods section

- Added as suggested (Page Number 20)

- please provide the Accession number for your deposited RNA seq data in Data Availability section

- Added as suggested, (Page Number 30)

Figure Check:

-Figure S5 A and B: is the 2nd row a duplicate? The background in the column 2 (A) and the column 3 (B) looks the same. Please provide source data for these two panels.

- Thank you for pointing this out. We really appreciate it. There has been a mix-up in the figures of the two panels. We have now corrected it and source files are also attached below.

Source Figure S5A-S5B

We also take this opportunity to thank the two reviewers of our original manuscript for their critical and insightful comments that help us to tremendously improve the manuscript.

Reviewer #1 (Comments to the Authors (Required)):

The authors have done an excellent job of addressing the reviewers' concerns, and in particular my request for better biochemical evidence that phosphorylation of Tyr202 in PAPI by Wzc is important for its stress induced inactivation. They have included a large number of new experiments to demonstrate that PAPI is phosphorylated on Tyr202 by Wzc under stress conditions and that this inhibits its activity, which normally polyadenylates stress response mRNAs leading to their destabilization. The one thing they were not able to do was to establish the stoichiometry of phosphorylation of Tyr202 in PAPI, which needs to be high for strong downregulation of PAPI polyadenylation activity under stress conditions - this could be determined by quantitative MS analysis of the pY202 and Y202 peptides from PAPI isolated from stressed and unstressed cells. However, it is not necessary for this experiment to be done before publication.

- We thank the reviewer for the insightful review that help us to improve our manuscript.

Points: 1. Figure 3P: These data are not very impressive, because the WT pY202 PAPI signal is so low.

- While we agree with the reviewer that signal in the two first two lanes is little low, the blot clearly shows detection of PY202 phosphorylation in wild type cells that is absent in the mutant cells.

2. Figure 4A/B: It is no unreasonable that a phosphate on Y202 will cause a conformational change in the ATP-binding pocket. However, while the structural simulation analysis suggests that Y202 phosphorylation might prevent ATP association, it does not prove this. In consequence, the authors should be careful about their conclusions, unless they carry out ATP binding studies on unphosphorylated and pY202 PAPI to demonstrate pY202 PAPI has decreased affinity for ATP.

- As suggested by the reviewer, we have now toned down statements of structural changes induced phosphorylation and its effect on ATP binding obtained from our simulation studies to reflect our actual data on likely effect of Y202 phosphorylation on the ATP incorporation into the PAP enzyme. For example, discussion, page 20 now reads “..... alterations may disrupt the catalytic active site affecting ATP binding leading to reduced polyadenylation.....”

3. Figures 3P/4G: From these two panels, it almost appears as though the Y314F PAPI mutation may promote Y202 phosphorylation - do the authors have any insights?

- It is unlikely. We have carried out these blots multiple times, however, the trend of these inductions are not consistent including both in qRT-PCR analysis and/or cellular viability experiments under stress, and hence we cannot conclude on this aspect.

4. *Figures 4M/S5E: The authors showed that Wzc RNA levels were increased by stress but apparently did not show that Wzc protein level is upregulated, which is the mechanism they are proposing for how PAPI Y202 phosphorylation is increased during stress. In Figure S5E, it appears that the same level of FLAG-Wzc protein was present in the stressed and unstressed cells, but its activity towards PAPI Y202 was strongly increased. Since Wzc phosphorylation would normally be regarded as a measure of its enzyme activity, it is somewhat surprising that Wzc autophosphorylation was not increased by stress when PAPI phosphorylation was strongly elevated. How do the authors explain this?*

- These experiments were carried out by transformation of the FLAG-Wzc expressing plasmid that is driven from an engineered constitutive promoter (stress independent, detailed in materials and methods) in MG1655 cells. Hence, the ectopically expressed Wzc protein detected using antibody against FLAG epitope did not show any change in the expression pattern. The induction of Y202 phosphorylation is likely from the induction in the endogenous Wzc present in the cell (that is evident from our qRT-PCR analysis). In the absence of anti-serum against Wzc protein, we cannot conclude on endogenous Wzc protein level inside the cell.

To address this concern, we have also transformed pFLAG^B-wzc construct in both wild type and wzc-mutant background. As expected in the absence of endogenous Wzc, there was neither induction in the P^{Y202} phosphorylation nor the FLAG-Wzc protein when transformed in a wzc-mutant background after stress treatment (Rebuttal Figure 1, Right panel, below).

Rebuttal Figure 1

5. *A model figure incorporating their findings/conclusions would help the reader.*

- As suggested by the reviewer, we have now incorporated a model depicting how Y202 PAPI phosphorylation by Wzc regulates stress response in bacteria in Figure S6.

Reviewer #2 (Comments to the Authors (Required):

The authors have carefully addressed all issues raised in my previous review. By new experiments and by text amendments the manuscript has been greatly improved. Notably, the authors generated a site-specific antibody against PAPI pTyr202, and key experiments with this antibody showed that tyrosine phosphorylation occurs as proposed before, and has a regulatory function. Moreover, the antibody enabled the authors to show

tyrosine phosphorylation of endogenous PAPI. Thereby all previously remaining concerns about the identity of the phosphorylated protein were resolved. All other points were also addressed by new experiments or improvements of the text. Therefore, I strongly recommend publication of this manuscript in its present form.

- We thank the reviewer for critically examining our manuscript that helped us to tremendously improve our manuscript.

November 29, 2022

RE: Life Science Alliance Manuscript #LSA-2021-01148-TRR

Dr. Rakesh S. Laishram
Rajiv Gandhi Centre for Biotechnology
Cancer Research program
Thycaud Post, Poojappura
Trivandrum 695014
India

Dear Dr. Laishram,

Thank you for submitting your Research Article entitled "Tyrosine phosphorylation controlled poly(A)polymerase activity regulates stress response in bacteria". It is a pleasure to let you know that your manuscript is now accepted for publication in Life Science Alliance. Congratulations on this interesting work.

DISTRIBUTION OF MATERIALS:

Again, congratulations on a very nice paper. I hope you found the review process to be constructive and are pleased with how the manuscript was handled editorially. We look forward to future exciting submissions from your lab.

Sincerely,
